# A single C-terminal residue controls SARS-CoV-2 spike trafficking and incorporation into VLPs

Debajit Dey[1], Enya Qing [2], Yanan He[3], Yihong Chen[3], Benjamin Jennings [4], Whitaker Cohn[5], Suruchi Singh[1], Lokesh Gakhar[6,7,15], Nicholas J. Schnicker [7,8], Brian G. Pierce [3,9], Julian P. Whitelegge [5,10,11], Balraj Doray [4], John Orban[3,12], Tom Gallagher [2] & S. Saif Hasan [1,13,14] ✉

The spike (S) protein of SARS-CoV-2 is delivered to the virion assembly site in the ER-Golgi Intermediate Compartment (ERGIC) from both the ER and cis-Golgi in infected cells. However, the relevance and modulatory mechanism of this bidirectional trafficking are unclear. Here, using structure-function analyses, we show that S incorporation into virus-like particles (VLP) and VLP fusogenicity are determined by coatomer-dependent S delivery from the cis-Golgi and restricted by S-coatomer dissociation. Although S mimicry of the host coatomer-binding dibasic motif ensures retrograde trafficking to the ERGIC, avoidance of the host-like C-terminal acidic residue is critical for S-coatomer dissociation and therefore incorporation into virions or export for cell-cell fusion. Because this C-terminal residue is the key determinant of SARS-CoV-2 assembly and fusogenicity, our work provides a framework for the export of S protein encoded in genetic vaccines for surface display and immune activation.

COVID-19 is caused by the coronavirus SARS-CoV-2, one of the most thoroughly studied virus to date, which is targeted by multiple different vaccines[1]. The SARS-CoV-2 virion comprises genomic RNA complexed with cytoplasmic nucleocapsid (N), which is surrounded by a lipid membrane containing S, membrane (M), and envelope (E) proteins[2–6]. Following infection of host cells, SARS-CoV-2 replicates and traffics its components to the ERGIC, where virions are assembled for subsequent release from the cell[3]. Viral assembly can occur in the absence of the S protein, but incorporation of S is required for the virion to bind, enter, and infect new host cells[7–9].

CoV assembly co-opts anterograde and retrograde secretory trafficking in infected cells[10–13]. In normal metabolism, anterograde

[1]Department of Biochemistry and Molecular Biology, University of Maryland School of Medicine, Baltimore, MD 21201, USA. [2]Department of Microbiology and Immunology, Loyola University Chicago, Maywood, IL 60153, USA. [3]University of Maryland Institute for Bioscience and Biotechnology Research, Rockville, MD 20850, USA. [4]Department of Internal Medicine, Hematology Division, Washington University School of Medicine, St. Louis, MO 63110, USA. [5]Pasarow Mass Spectrometry Laboratory, The Jane and Terry Semel Institute for Neuroscience and Human Behavior, David Geffen School of Medicine, University of California, Los Angeles, CA 90095, USA. [6]Department of Biochemistry and Molecular Biology, Carver College of Medicine, University of Iowa, Iowa City, IA 52242, USA. [7]Protein and Crystallography Facility, Carver College of Medicine, University of Iowa, Iowa City, IA 52242, USA. [8]Department of Molecular Physiology and Biophysics, Carver College of Medicine, University of Iowa, Iowa City, IA 52242, USA. [9]Department of Cell Biology and Molecular Genetics, University of Maryland, College Park, MD 20742, USA. [10]Molecular Biology Institute, University of California, Los Angeles, Los Angeles, CA 90095, USA. [11]Jonsson Comprehensive Cancer Center, University of California, Los Angeles, Los Angeles, CA 90095, USA. [12]Department of Chemistry and Biochemistry, University of Maryland, College Park, MD 20742, USA. [13]University of Maryland Marlene and Stewart Greenebaum Cancer Center, University of Maryland Medical Center, Baltimore, MD 21201, USA. [14]Center for Biomolecular Therapeutics, University of Maryland School of Medicine, Rockville, MD 20850, USA. [15]Present address: PAQ Therapeutics, Burlington, MA 01803, USA. ✉e-mail: sshasan@som.umaryland.edu

trafficking is responsible for the secretory export of endogenous cargo proteins out of ER[14–20] (Fig. 1A). This export of cargo molecules is accompanied by the exodus of ER-resident proteins such as type I membrane proteins that are critical for ER-specific metabolic functions[21–24]. Hence, this loss of critical proteins has the potential to compromise ER proteostasis and generate cellular stress. This

challenge to normal metabolism is overcome by the activity of the cytosolic coatomer protein I (COPI) polymer, whose repeating unit is the coatomer, a 560 kDa complex of seven distinct gene products (α-, β-, β′-, γ-, δ-, ε-, and ζ-COPI)[25–31] (Fig. 1A). The α, β′, and ε subunits constitute the "B-subcomplex" whereas the β, δ, γ, and ζ subunits constitute the "F-subcomplex", which shares significant structural

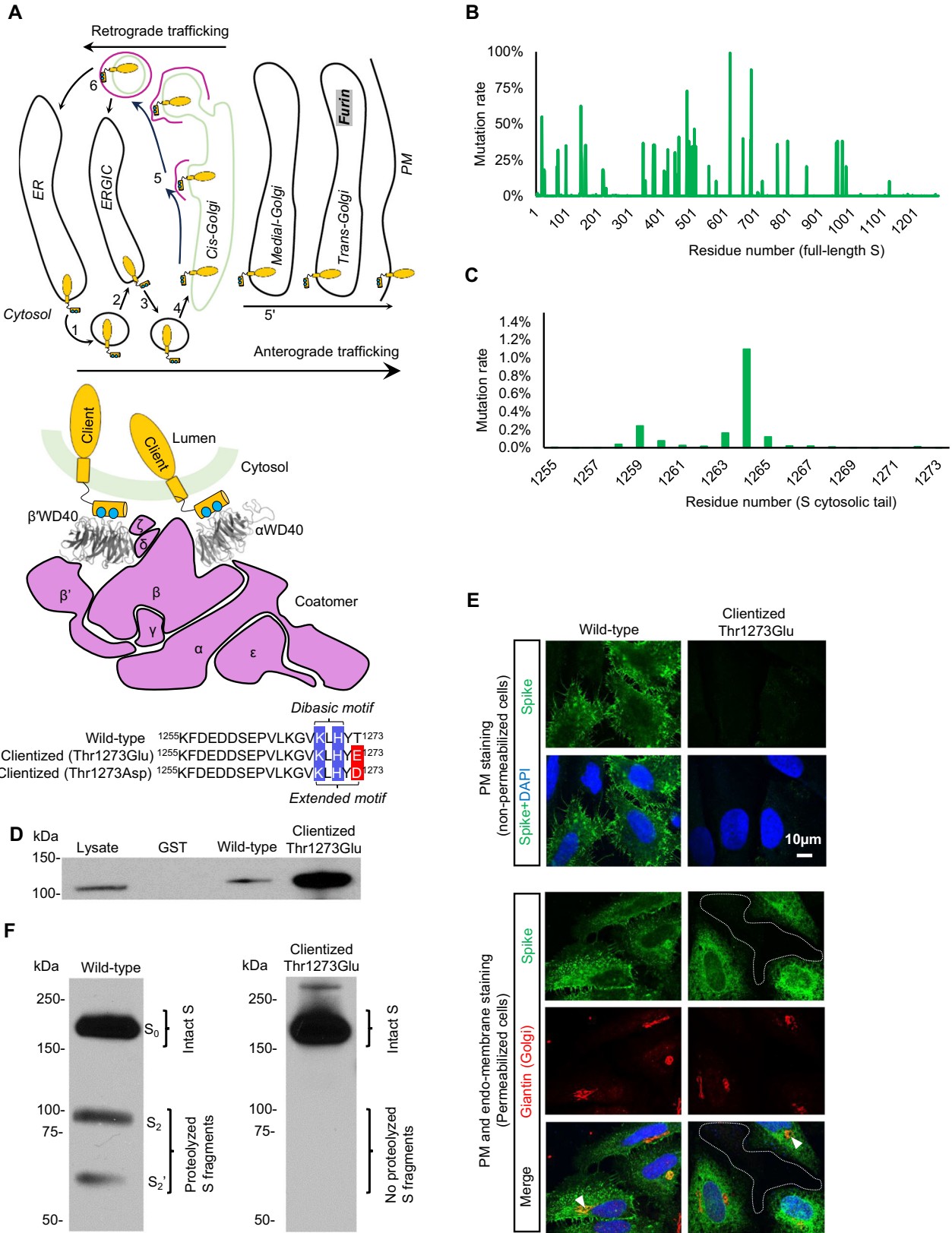

**Fig. 1 | The conserved C-terminus determines S protein trafficking and coatomer interactions. A** Upper panel: Schematic of bidirectional trafficking of S in the secretory pathway. The newly synthesized S protein (yellow) in the ER (endoplasmic reticulum) is packaged in vesicles and trafficked to the ERGIC (endoplasmic reticulum-Golgi intermediate complex; steps 1, 2), and delivered to the cis-Golgi (steps 3, 4). COPI-coated vesicles (pink) retrieve S to the ER and ERGIC by retrograde trafficking (steps 4-6). S not retrieved by COPI is exported to the PM (step 5′) via the furin-containing trans-Golgi by anterograde trafficking. Middle panel: The coatomer is a hetero-heptamer of seven distinct gene products. N-terminal WD40 domains on α and β′ subunits form the binding site for client proteins such as S. The illustration in panel A was created using Microsoft PowerPoint. Lower panel: Alignment of S residues in the extra-membrane domain of the cytosolic tail (1255-1273) showing the dibasic motif, extended motif, and clientizing substitutions at the C-terminus. **B** Multiple mutational hotspots in the N-terminal two-thirds of the full-length SARS-CoV-2 S, which corresponds to the ectodomain. **C** Few mutations occur in the S tail. **D** Clientized GST-S tail fusion protein pulls down ~17-fold more coatomer than wild-type S tail-GST fusion. This represents one of three biologically independent samples. **E** Immunofluorescence imaging of full-length wild-type and clientized S in permeabilized and non-permeabilized HeLa cells. Wild-type S localizes to the PM and early secretory compartments. Clientized S protein is absent from the PM and restricted to early secretory compartments. White outline, intra- and inter-cellular space devoid of S; white arrowheads, early secretory compartments. This represents one of two biologically independent samples. **F** Western blots of HeLa cells expressing wild-type or clientized full-length S. Wild-type S yields intact, unproteolyzed fractions from early secretory compartments and proteolyzed, furin-cleaved fragments from PM (left). Clientized S yields only an intact fraction from early secretory compartments (right). $S_0$: intact S; $S_2$, $S_2'$: proteolyzed fragments. This represents one of two biologically independent samples.

similarity with the clathrin adaptors that function in PM trafficking[31–33]. Overall, the coatomer functions to retrieve the escaped "client" proteins, such as type I membrane proteins, from cis-Golgi back to ER by retrograde trafficking. The coatomer binds to a C-terminal dibasic retrieval motif, Lys-x-Lys-x-$x_{CT}$ or Lys-Lys-x-$x_{CT}$ (x = any amino acid; $x_{CT}$ = any C-terminal amino acid) in the cytosolic tail of type I membrane protein clients, which leads to client packaging into COPI-coated vesicles and retrieval to ER[34–40]. The N-terminal β-propellor WD40 domains of B-subcomplex subunits, α and β′ (αWD40 and β′WD40, respectively) provide binding sites for the dibasic motif on client proteins[37,41–43]. Since the coatomer functions to retrieve a wide variety of eukaryotic proteins, it is not surprising that coatomer dysfunction has broad secretory effects and is linked to multiple disorders of growth, development, auto-immunity, and cancers[24,43–54]. SARS-CoV-2 hijacks this secretory system to assemble its components into virions. In particular, the S protein – a type I membrane protein – is delivered to the ERGIC from both the ER (by anterograde trafficking) and the Golgi (by retrograde trafficking)[3,10–12] (Fig. 1A). Retrograde delivery of S is due to a Lys-x-His-x-$x_{CT}$ sequence in its cytosolic tail, which partially mimics the coatomer-binding dibasic motif[11,12]. However, $x_{CT}$ is frequently an acidic residue in endogenous client proteins and strictly non-acidic in CoV S molecules[43], revealing an abrupt departure from mimicry immediately downstream of the dibasic motif (S protein extended motif). The significance of this bipartite tail organization and how it modulates incorporation of the S protein into virions is unclear. Moreover, little is known about the relevance of bidirectional S trafficking during virion assembly.

Here we utilize recombinant SARS-CoV-2 S proteins and tail peptides, including constructs with enhanced mimicry of the coatomer binding motif (clientized S protein), to elucidate the atomic structure of the S-coatomer interface. We provide functional insight into the unconventional single-residue modification of the C-terminal retrieval motif, revealing its role in S-coatomer dissociation and incorporation of S into virions. Finally, we reveal the importance of retrograde trafficking for fusogenicity and propagation of SARS-CoV-2.

## Results

### The conserved C-terminus determines S protein trafficking and coatomer interactions

To determine whether residues in the SARS-CoV-2 S tail are under selection pressure, we analyzed over 11 million sequences of full-length SARS-CoV-2 S in the GISAID database[55,56] (Fig. 1B). Residues predicted to make substantial contact with the coatomer surface (Lys1269, His1271, and Thr1273) showed minimal mutation rates (0.0020%, 0.0015%, and 0.0004%, respectively) (Fig. 1C). The highest frequency mutation at the S protein C-terminus was Thr1273Ser. Importantly, mutations predicted to increase mimicry by the S protein extended motif (Thr1273Asp or Thr1273Glu), were not reported.

We subsequently asked whether such clientizing mutations modify coatomer interactions in vivo and localization in secretory compartments. We first employed a pull-down assay of the endogenous coatomer complex with an N-terminal glutathione S-transferase (GST) fusion of the S tail (Leu1244 to Thr1273). A list of all the S constructs used in this investigation is provided in Supplementary Table S1. Clientized GST-S tail constructs containing either a Thr1273Asp or Thr1273Glu substitution showed 17-fold higher affinity for the β-COPI subunit (a marker for the intact coatomer complex) than the wild-type GST-S tail construct (Fig. 1D; Supplementary Fig. S1). Furthermore, these clientizing mutations enhanced binding by 2-fold compared to the canonical S tail Lys-x-Lys-x-$x_{CT}$ dibasic mutant. Mass spectrometry identified all seven coatomer subunits in pull-downs with GST-S tail clientized constructs (Supplementary Table S2). These data show that clientizing mutations outside the dibasic motif enhance coatomer binding affinity compared to the endogenous canonical Lys-x-Lys-x-$x_{CT}$ motif, suggesting that the S $x_{CT}$ residue has a key modulatory function in coatomer interactions.

Given that the coatomer mainly associates with early secretory compartments, we next evaluated whether enhanced coatomer association due to a clientizing substitution alters the localization of S proteins. Immunofluorescence microscopy showed full-length wild-type S both at the PM and in secretory compartments (Fig. 1E). In contrast, clientized S carrying the Thr1273Glu mutation became sequestered in early secretory compartments and did not reach the PM (Fig. 1E). We next asked whether relocalization of clientized S was due to lack of export from early secretory compartments or recycling from the PM (Fig. 1A). Cleavage of S protein into S1 and S2 fragments by the trans-Golgi protease furin is a marker for trafficking out of early secretory compartments[57–60] (Fig. 1A). We observed both an intact band and proteolyzed fragments for wild-type S, corresponding to secretory compartment- and PM-localized populations, respectively (Fig. 1F). In contrast, a single, intact band was observed for clientized Thr1273Glu S protein, corresponding to retention in early secretory compartments rather than recycling from the PM (Fig. 1F). Together, these data demonstrate that the C-terminus, immediately downstream of the dibasic motif, is a critical modulator of S trafficking.

### The clientized S protein C-terminus engages a basic cluster in WD40 domains

The SARS-CoV-2 tail displays selectivity for αWD40 over β′WD40[43]. To determine an atomic-level understanding of S-coatomer interactions, we attempted co-crystallization of αWD40 with either wild-type (1267Gly-Val-Lys-Leu-His-Tyr-Thr1273) or clientized (1267Gly-Val-Lys-Leu-His-Tyr-Glu1273) S tail heptapeptides. We used WD40 domains from the yeast α-COPI and β′-COPI subunits, which have been used extensively in prior structural investigations[43,61,62]. The binding site residues are 100% identical between yeast and human αWD40

domains (PDB IDs 7S22 and 6PBG). The yeast and human β′WD40 domain binding sites differ at one residue, i.e., Phe142 in yeast β′WD40 is replaced by Tyr144 in human β′WD40 (PDB ID 8D30). However, in our co-crystallization experiments, contacts between symmetry-related αWD40 molecules occluded the binding site and no S tail peptide density was observed. Hence, we subsequently adopted a previously described strategy of using β′WD40 for co-crystallization of peptides containing the Lys-x-His-x-x$_{CT}$ motif [61], which was also used for structural investigations of S tail peptides[63], recently during the revision of this manuscript. The yeast αWD40 and β′WD40 domains are structural homologs with complete conservation of the binding site except for two residues, i.e., His31/Tyr139 in αWD40 are replaced by Tyr33/Phe142 in β′WD40[61,62] (Supplementary Fig. S2). The β′WD40 co-crystal structures with the wild-type and clientized Thr127Glu S heptapeptides were determined to a resolution of 1.4 Å (Supplementary Table S3). Overall, these two crystal structures reveal substantial similarity (Cα RMSD = 0.2 Å). The S tail heptapeptides are intercalated between two stacked β′WD40 domains. The S tail extended motif makes contact with the binding site residues in one of the two β′WD40 domains whereas the upstream tail residues interact with the other β′WD40 domain, but at the face distal from the binding site (Supplementary Fig. S3). The major difference is that whereas only the main chain carboxylate of wild-type Thr1273 interacts with basic Lys17 on the β′WD40 surface, both the side chain and main chain carboxylate groups of the clientized Glu1273 residue form extensive interactions with a β′WD40 basic cluster of Arg15, Lys17, and Arg272 (Fig. 2A, B). This suggests that interactions between Glu1273 and the basic cluster underlie the enhanced binding of the clientized S protein. Overall, the two heptapeptides occupy the canonical WD40 binding site, which provides electrostatic complementarity to the dibasic motif residues, Lys1269 and His1271 (Fig. 2C; Supplementary Fig. S4A, B). Furthermore, the S tail heptapeptides demonstrate substantial structural similarity with previously published co-crystal structures of β′WD40 domain with the tail peptides from endogenous client proteins and a porcine CoV S protein[61,62] (Supplementary Table S5).

To gain insight into the role of the basic cluster in stabilizing S-coatomer interactions, we generated alanine mutants of αWD40 Arg13, Lys15, and Arg300, which are equivalent to the β′WD40 basic cluster residues Arg15, Lys17, and Arg272, respectively. Biolayer interferometry (BLI) assays showed that Arg13Ala weakened αWD40 binding to both wild-type and clientized Thr1273Glu S heptapeptides, although the affinity for the clientized S heptapeptide was still 2.8-fold higher (Table 1, Fig. 2D, E). A 1.9 Å resolution crystal structure of this αWD40 Arg13Ala mutant showed a binding site architecture comparable to that in wild-type αWD40 (Fig. 2F; Supplementary Fig. S4C; Supplementary Table S4), attributing affinity weakening to the alanine mutation. The αWD40 Arg300Ala mutation yielded similar dissociation constants ($K_D$) of 3.6 ± 0.4 μM and 4.2 ± 0.4 μM for wild-type and clientized Thr1273Glu S heptapeptides, respectively (Table 1, Fig. 2G, H), and a 1.8 Å resolution crystal structure showed minimal perturbation of the binding site in this β′WD40 mutant (Fig. 2I; Supplementary Fig. S4D; Supplementary Table S3). This suggests that the αWD40 Arg300 side chain is important for the higher affinity for the clientized S heptapeptide. Finally, the Lys15Ala mutant did not bind to either the wild-type or clientized Thr1273Glu S heptapeptides (Fig. 2J, K), and a 1.6 Å resolution crystal structure revealed this was due to conformational occlusion of the αWD40 binding site by an inward rotation of nearby Arg13 and His31 side chains, rather than loss of Lys15 interactions (Fig. 2L; Supplementary Fig. S4E; Supplementary Table S4). We therefore generated wild-type and clientized Thr1273Glu S heptapeptides with amidation at the C-terminal main chain carboxylate, which neutralizes the main chain charge that interacts with αWD40 Lys15. This approach does not perturb the binding site architecture of αWD40, unlike mutagenesis. Loss of αWD40 binding was observed for the amidated wild-type S heptapeptide (Fig. 2M, Table 1) but not the

amidated clientized Thr1273Glu S heptapeptide (Fig. 2N, Table 1). Hence, the Glu1273 side chain in the clientized S protein provides partial compensation for the loss of main chain stabilization from the WD40 binding site, confirming the importance of this acidic residue for WD40 domain binding.

## The clientized S tail has broad selectivity for αWD40 and β′WD40

Prior work revealed that the wild-type S heptapeptide does not interact strongly with β′WD40[43], even though there is complete conservation of the basic cluster in αWD40 and β′WD40 domains[61,62]. In fact, the αWD40 domain is favored by several client tails over the β′WD40 domain (Supplementary Table S6). We, therefore, asked whether clientization of the S heptapeptide enables binding to β′WD40. Indeed, clientized Thr1273Glu S heptapeptide demonstrated binding to purified β′WD40 with a $K_D$ of 2.30 ± 0.23 μM (Table 1, Fig. 3A, B). This suggests that the x$_{CT}$ residue, which is outside the dibasic motif, is a critical determinant of S-coatomer selectivity. Furthermore, our co-crystal structures showed that the Tyr33 side chain of β′WD40 pushes away the Tyr1272 side chain of the S heptapeptide (Fig. 3C). This Tyr33 is replaced by a smaller His31 residue in the αWD40 domain (Supplementary Fig. S2). We hypothesized that the replacement of the bulky Tyr33 side chain with a smaller residue would allow Tyr1272 to move closer to the β′WD40 surface, thereby further strengthening the binding between S and β′WD40. Therefore, we generated a β′WD40 Tyr33Ala mutant and determined its co-crystal structure with the clientized S heptapeptide at a resolution of 1.8 Å (Supplementary Table S3). Although the S Tyr1272 side chain was indeed closer to the mutant β′WD40 surface by 0.8 Å (Fig. 3D; Supplementary Fig. S4F), overall, the Tyr1272 side chain was still >4 Å away from the β′WD40 surface. A BLI assay revealed this was insufficient to enhance S-β′WD40 affinity (Fig. 3E). Moreover, this β′WD40 Tyr33Ala mutation abolished binding to the clientized Thr1273Glu S heptapeptide (Fig. 3F), suggesting a previously unrecognized role for β′WD40 Tyr33 in S heptapeptide binding.

The enhanced interaction of the clientized Thr1273 Glu S tail heptapeptide provided us with a tool to test the validity of using β′WD40 domain in our structural experiments. We hypothesized that if the interaction site is comparable between the αWD40 and β′WD40 domains, then a disruptive substitution at a juxtaposed critical WD40 residue should yield a similar weakening of S tail interaction with either domain. Therefore, we expressed and purified an alanine substitution mutant of β′WD40 Lys17, which is juxtaposed with αWD40 Lys15. This β′WD40 mutant demonstrated a lack of binding to the clientized Thr1273Glu S tail heptapeptide, which is consistent with the loss of binding displayed by the αWD40 Lys15Ala mutant (Fig. 3G). Next, we determined a 2 Å resolution crystal structure of this domain (Supplementary Table S4; Supplementary Fig. S4G). This showed a rotation of the Arg15 side-chain reminiscent of the conformational change reported above in the αWD40 Lys15Ala mutant (Fig. 3H; Supplementary Fig. S5). These experiments show an atomic and residue-level equivalence in the binding interactions offered by the αWD40 and β′WD40 domains to the S tail, thus validating the use of the β′WD40 domain in structural investigations.

## Clientization alters the local and global conformations of S tail

Because the clientizing mutation Thr1273Glu generates a side chain and main chain anionic cluster in a highly charged S tail (Supplementary Table S1), we sought to determine whether clientization induces a distinct conformation. We first obtained backbone resonance assignments for a 21 residue,$^{13}$C/$^{15}$N-labeled wild-type S tail construct that includes the entire extra-membrane segment of the S protein, from Lys1255 to Thr1273 (Fig. 4A; Supplementary Table S1;). We then assigned the corresponding Thr1273Glu mutant S tail peptide. Comparison of $^{13}$Cα and $^{13}$CO secondary chemical shifts indicates weak

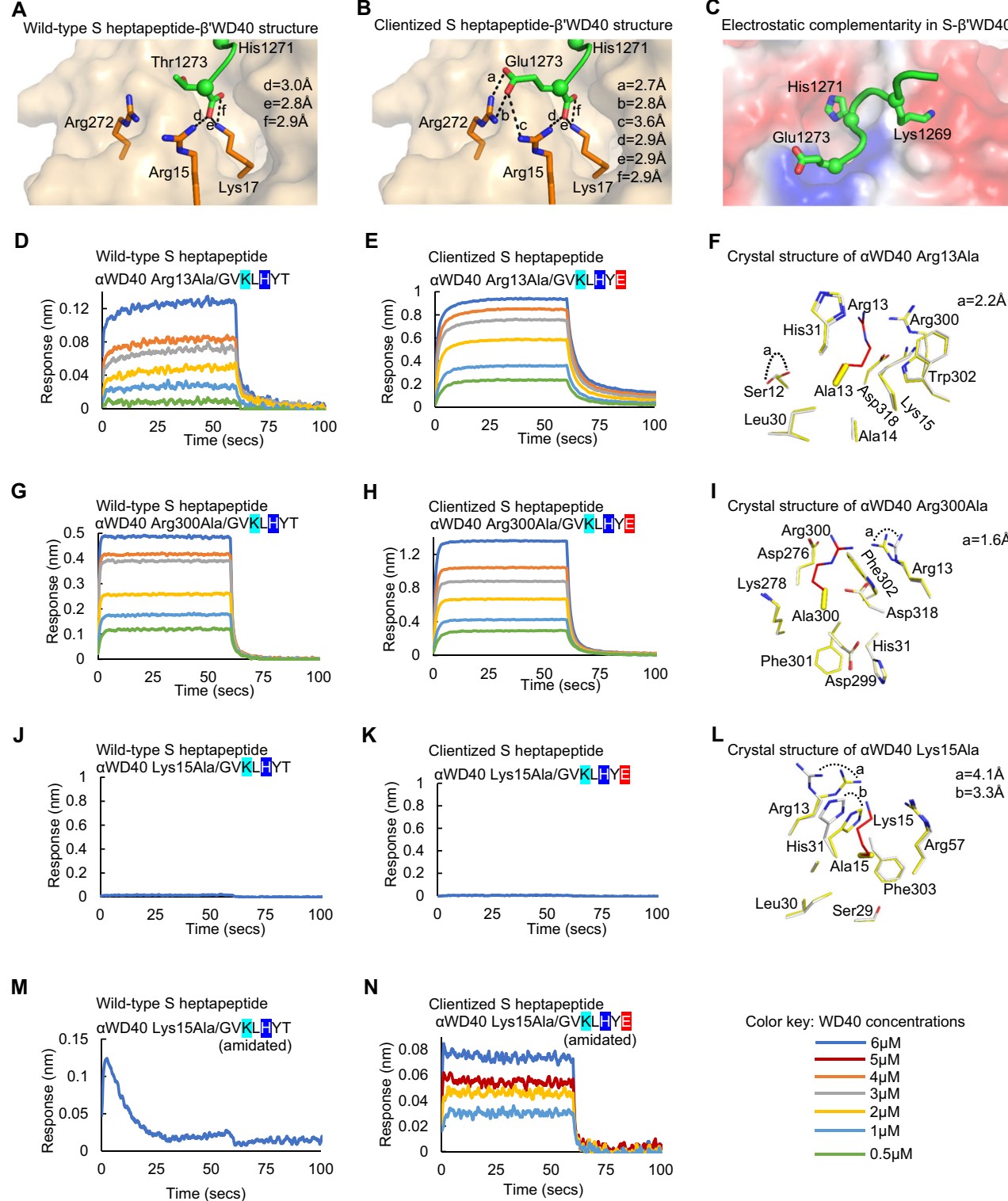

**Fig. 2 | Clientized S binds to a basic cluster on the coatomer WD40 domain.**
**A**−**C** Co-crystal structures of SARS-CoV-2 S tail heptapeptides (residues 1267-1273) in complex with β'WD40. **A** Thr1273 in wild-type S shows no side-chain interactions with the basic cluster. **B** Glu1273 in clientized S forms complementary electrostatic interactions with the β'WD40 basic cluster. **C** The β'WD40 surface provides electrostatic complementarity to the S dibasic motif and the C-terminus. Color code: red, acidic; blue, basic. **D**, **E** BLI assays show that αWD40 Arg13Ala mutant has weaker affinity for **D** wild-type S heptapeptide than for, **E** Thr1273Glu clientized S heptapeptide. **F** Superposition of wild-type (white) and Arg13Ala mutant (yellow) crystal structures shows minimal perturbation near the mutation site. **G**, **H** BLI assays show that αWD40 Arg300Ala mutant has similar affinity for, **G** wild-type S

heptapeptide and, **H** Thr1273Glu clientized S heptapeptide. **I** Superposition of wild-type (white) and Arg300Ala mutant (yellow) crystal structures shows minimal perturbation near the mutation site. **J**, **K** BLI assays show that αWD40 Lys15Ala mutant shows no interaction with,(**J** wild-type S heptapeptide or, **K** Thr1273Glu clientized S heptapeptide. **L** Superposition of wild-type (white) and Lys15Ala mutant (yellow) crystal structures shows substantial perturbation and inward rotation of Arg13 and His31 side chains in the mutant to block the S heptapeptide binding site. **M**, **N** BLI assays for αWD40 shows loss of binding with, **M** amidated wild-type S heptapeptide but not, **N** amidated Thr1273Glu clientized S heptapeptide. Color key indicates αWD40 concentrations for BLI assays.

**Table 1 | Binding rates and dissociation constants of S tail peptides interacting with coatomer WD40 domains[1]**

| Peptide sequence | WD40 analyte | Equilibrium KD (µM) | $k_{on}$ ($10^3$/Ms) | $k_{off}$ ($10^{-1}$/s) |
|---|---|---|---|---|
| GVKLHYT[2] | α | 1.40 ± 0.15 | 152.13 ± 3.45 | 1.45 ± 0.01 |
| GVKLHYT | β' | n.d. | n.d. | n.d. |
| GVKLHYE[2] | α | 0.31 ± 0.01 | 58.96 ± 0.54 | 0.24 ± 0.003 |
| GVKLHYE | β' | 2.3 ± 0.23 | 2424.7 ± 178.3 | 12.7 ± 0.3 |
| GVKLHAT | β' | n.d. | n.d. | n.d. |
| KEVYLHG | β' | n.d. | n.d. | n.d. |
| GVKLHYT | β'-Tyr33Ala | n.d. | n.d. | n.d. |
| GVKLHYE | β'-Tyr33Ala | n.d. | n.d. | n.d. |
| GVKLHYT (amidated) | α | n.d. | n.d. | n.d. |
| GVKLHYE (amidated) | α | 2.3 ± 0.7 | 706.2 ± 56.3 | 8.8 ± 0.2 |
| KEVYLHG | α-Tyr139Ala | n.d. | n.d. | n.d. |
| GVKLHYT | α-Arg13Ala | 8.5 ± 2.0 | 147.7 ± 7.0 | 3.8 ± 0.07 |
| GVKLHYE | α-Arg13Ala | 3.0 ± 0.2 | 132.5 ± 4.5 | 2.2 ± 0.08 |
| GVKLHYT | α-Arg300Ala | 3.6 ± 0.4 | 360.1 ± 9.4 | 8.1 ± 0.07 |
| GVKLHYE | α-Arg300Ala | 4.2 ± 0.4 | 136.6 ± 2.6 | 4.3 ± 0.03 |
| GVKLHYT | α-Lys15Ala | n.d. | n.d. | n.d. |
| GVKLHYE | α-Lys15Ala | n.d. | n.d. | n.d. |
| GVKLHYE | β'-Lys17Ala | n.d. | n.d. | n.d. |
| CCKFDEDDSEPVLKGVKLHYT | α | 1.7 ± 0.083 | 99.3 ± 1.5 | 2.67 ± 0.01 |
| CCKFDEDDSEPVLKGVKLHYE | α | 0.1 ± 0.005 | 59.9 ± 1.3 | 0.037 ± .002 |
| CCKFDEDDSEPVLKGVKLHYT[3] | α | 1.1 ± 0.1 | 145.5 ± 4.4 | 2.9 ± 0.03 |
| CCKFDEDDSEPVLKGVKLHYE[3] | α | 0.1 ± 0.002 | 60.8 ± 1.3 | 0.03 ± 0.002 |

[1]$\chi^2$ values < 3 for all assays.
[2]Reference: [43]Dey, D. et al. An extended motif in the SARS-CoV-2 spike modulates the binding and release of host coatomer in retrograde trafficking. Commun Biol 5, 115, doi:10.1038/s42003-022-03063-y (2022).
[3]These assays were performed at pH 7.0 as opposed to pH 6.5 for other assays.

consensus β-strand propensity (consecutive negative ΔCα and ΔCO values) for N-terminal half residues Glu1258-Asp1260 in both the wild-type and mutant S tails (Fig. 4B). In contrast, the C-terminal half residues Val1268-His1271, which include the dibasic motif Lys1269-His1271, have consensus β-propensity in the wild-type S tail but not the Thr1273Glu clientized mutant. These data indicate that the Thr1273Glu mutation alters the conformational preferences of the WD40 binding residues in the S tail. This is further supported by the data in Supplementary Fig. S6, which shows that the Thr1273Glu mutation significantly affects the backbone amide chemical shift of Leu1270, located in the center of the S tail dibasic motif. Together, the NMR results are indicative of transient hydrogen bonding interactions between Glu1273 and Leu1270, which is at the center of the dibasic motif. This is suggestive of potential and likely transient interactions between acidic Glu1273 and the sequentially nearest (i-4) basic residue Lys1269. Such a proximal Lys1269-Glu1273 arrangement (even a transient one) would be expected to be less favorable for WD40 binding, given that the bound state of the S tail in our X-ray structures has Lys1269 and Thr/Glu1273 in a distal configuration. We tested this hypothesis using a comparative BLI analysis of wild-type and Thr1273Glu mutant of the S tail 21mer interactions with the αWD40 domain. Indeed, while the 21mer Thr1273Glu mutant S tail binds to WD40 domains more tightly than the 21mer wild-type S tail, its binding kinetics are substantially slower than of the wild-type S tail 21mer (Fig. 4C, D, Table 1).

Next, we directly evaluated conformational changes induced by the clientizing mutation in the full extra-membrane S tail 21mer (Lys1255-Thr/Glu1273) using paramagnetic relaxation enhancement (PRE)[64,65]. The spin label MTSL (S-(1-oxyl-2,2,5,5-tetramethyl-2,5-dihydro-1H-pyrrol-3-yl)methyl methanesulfonothioate) was conjugated to a single cysteine at position 1254 near the N-terminus of the 21mer S tail (Supplementary Table S1). Amide peak intensities in the 2D $^1H$-$^{15}N$ heteronuclear single quantum coherence (HSQC) spectrum of this $^{15}N$-isotope labeled and MTSL-conjugated S tail were measured in the oxidized ($I_{ox}$) and reduced ($I_{red}$) states (Fig. 4E). As expected, the 21mer S tail backbone amide protons of residues nearest in sequence to the spin label were paramagnetically broadened with decreased peak intensities in the oxidized state[64]. In addition, however, some of the backbone amide protons of residues Val1264-His1271, which are more removed from the spin label site also had significant PRE effects. This suggested a flexible hinge-like structure centered around Glu1262 and Pro1263 with transient contacts between the N- and C-terminal halves of the S tail. We note that the extent of these longer-range transient interactions is slightly, but consistently, attenuated in the clientized Thr1273Glu S tail (Fig. 4E), indicating that the clientizing mutation weakens transient contacts between the N- and C-terminal halves. Electrostatic interactions between an acidic region near the N-terminus (Asp1257-Asp1260) and basic residues Lys1266 and Lys1269 in the C-terminal half are the most likely drivers for these long-range transient contacts. When the C-terminal Thr1273 is mutated to Glu1273, this alters the charge balance, likely resulting in weakened interactions between the N- and C-terminal halves of the S tail. The PRE results are therefore consistent with the secondary chemical shift analysis (Fig. 4B) and chemical shift perturbation (CSP) data (Supplementary Fig. S6) described above, whereby the change in local conformational preferences induced by the mutation also contribute to the differences seen in the PRE profiles of the wild-type and Thr1273Glu mutant S tails.

Finally, we determined the relative binding of full-length, extra-membrane S tail 21mer to αWD40 and β'WD40 domains using NMR. Wild-type S tail CSPs were substantially smaller for β'WD40 than αWD40, indicating weaker binding to β'WD40, but clientized S tail CSPs revealed enhanced binding to β'WD40 (Supplementary Fig. S7). Furthermore, our NMR experiments on S 21mer constructs identified S

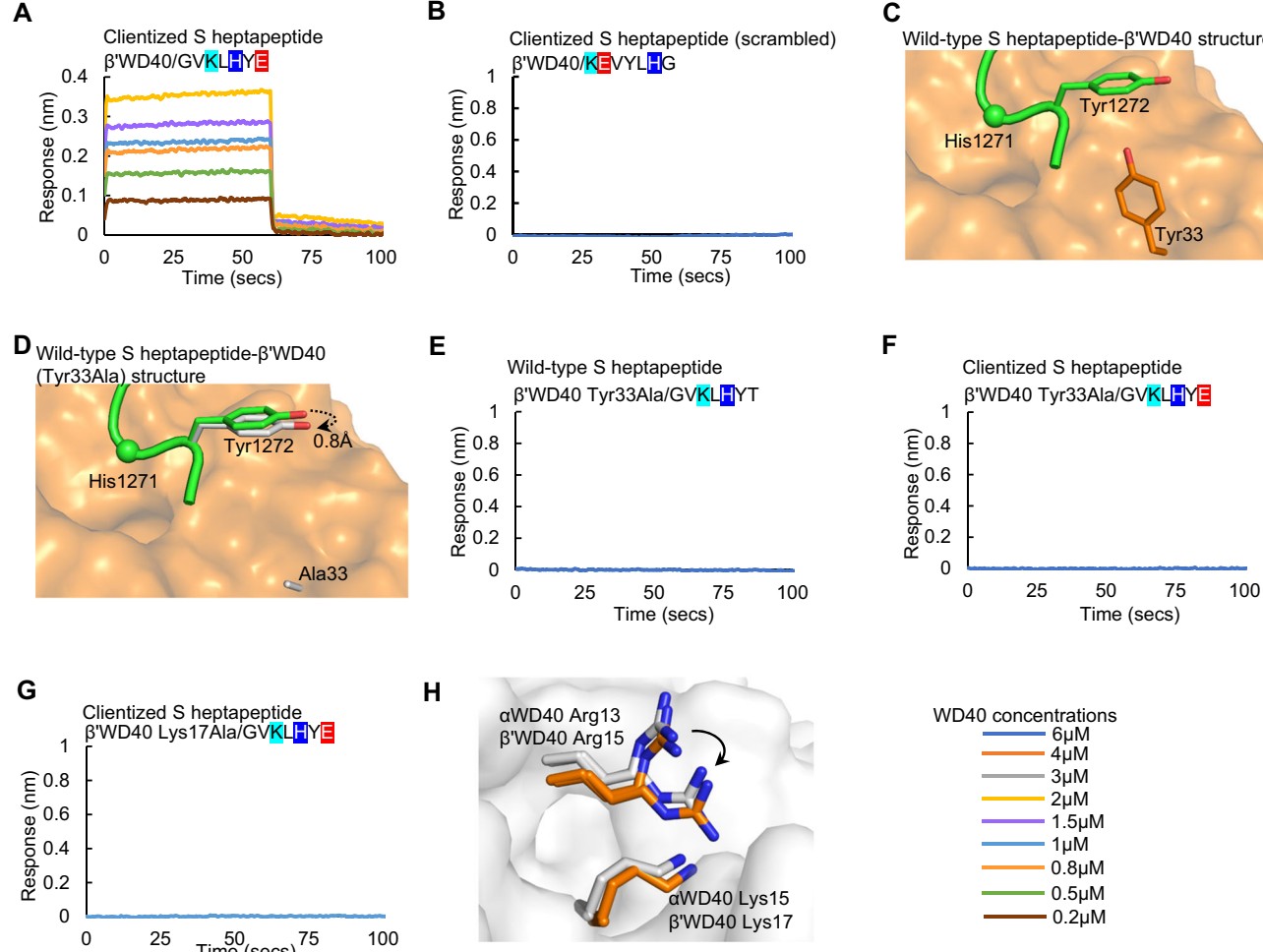

**Fig. 3 | The clientized S tail has broad selectivity for αWD40 and β′WD40 domains. A**, **B** BLI assays show direct binding of β′WD40 to, **A** Thr1273Glu clientized S heptapeptide but not to, **B** clientized S heptapeptide with a scrambled sequence. **C** Co-crystal structure shows the β′WD40 Tyr33 side chain pushing the Tyr1272 side chain in the Thr1273Glu clientized S heptapeptide away from the domain surface. **D** Co-crystal structure shows that substitution of the β′WD40 Tyr33 side chain for Ala causing minimal movement of the S tail Tyr1272 side chain towards the domain surface. **E**, **F** BLI assays show that β′WD40 Tyr33Ala mutant does not bind to either, **E** wild-type or **F** clientized S heptapeptide. **G** BLI assay shows loss of binding of β′WD40 Lys17Ala mutant to Thr1273Glu clientized S heptapeptide. **H** Superposition shows a similar conformational arrangement of juxtaposed Arg residues near the substitution sites in αWD40 Lys15Ala (white-blue) and β′WD40 Lys17Ala (orange-blue) crystal structures. Color key indicates WD40 concentrations for BLI assays.

tail residues well upstream the crystallographic footprint that demonstrate CSPs upon binding to αWD40 (Supplementary Fig. S7). These CSPs could be due to direct contacts between the S tail and WD40 domain, reflecting a larger binding footprint for the full-length S tail, or indirect effects such as a loss of long-range intra-tail contacts upon binding to αWD40. Interestingly, the upstream residues Ser1261-Glu1262, which modulate coatomer interactions[11], are in a turn that likely modulates N- and C-terminal interactions in the S tail. Nevertheless, these various data show that clientization enhances the binding of S proteins to WD40 domains.

### Enhanced coatomer affinity reduces S-directed membrane fusion

Having established that clientization of the S protein enhances coatomer binding and localization in early secretory compartments, we asked whether it might also modify export to the host cell PM as well as fusogenicity. To address these questions, we performed a dual split protein (DSP) assay in cell culture[66] (Fig. 5A). In this assay, the full-length, 1273 residue S protein is on the PM of "effector" cells, which express one-half of a split luciferase along with M, E, and N-HiBiT. The receptor ACE2 is on "target" cells, which express the other half of the

split luciferase. S-ACE2 interaction and subsequent fusion are followed by the reconstitution of the luciferase, whose activity serves as a readout for S protein fusogenicity. In contrast to wild-type S protein, which yielded a robust luciferase signal, clientized S protein containing either a Thr1273Glu or Thr1273Asp mutation showed no significant enhancement of luciferase activity (Fig. 5B). Moreover, western blot analysis of effector cell lysates revealed an intact, unproteolyzed population of clientized S proteins, revealing a lack of trafficking from early secretory compartments (Fig. 5C). This further suggests that M, E, and N structural proteins are unable to liberate clientized S from coatomer for export to the PM.

Next, we asked whether intracellular retention of clientized S increases its abundance at SARS-CoV-2 budding sites, conceivably promoting progeny infectivity, or whether strong coatomer association inhibits incorporation of S into assembling virions. In the latter scenario, CoV particles will have a lower abundance of S proteins and thus diminished cell entry potential. We employed a SARS-CoV-2 VLP assembly system that utilized luminescence as an assay for S-dependent entry of VLPs[66,67] (Fig. 5D). VLPs secreted from cells containing wild-type S protein generated robust luminescence and contained abundant furin-cleaved S (Fig. 5E, F). In

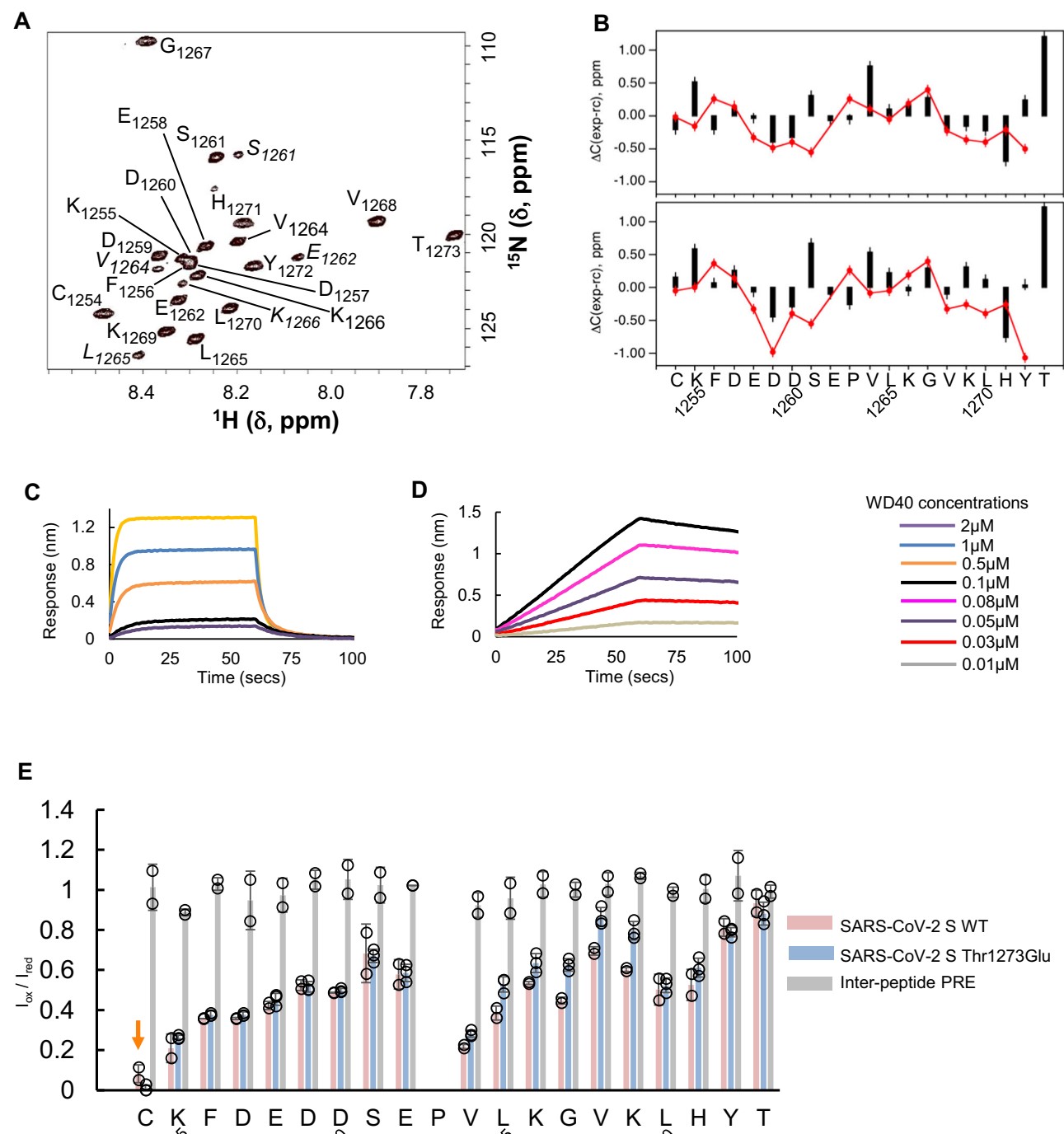

**Fig. 4 | Clientization alters the S tail conformation and strengthens binding to WD40 domains. A** Two-dimensional $^1H$-$^{15}N$ HSQC spectrum of wild-type S tail 21-mer peptide at 25°C with backbone amide assignments. Assignments in italics are putatively due to a small population (~10%) of peptide with cis-Pro1263. **B** Plot of $^{13}C$ secondary chemical shifts (experiment-calculated random coil) in the wild-type (top) and Cys1253Ala-Thr1273Glu clientized (bottom) S tail peptides for Cα (black) and CO (red) resonances. Weak consensus β-propensity is observed in residues Glu1258-Asp1260 for both the wild-type and clientized S-tails and in the dibasic motif-containing Val1268-His1271 for the wild-type S tail but not the clientized S tail. The error bars represent an estimate for the error in measuring the $^{13}C$ chemical shifts based on two independent experiments with wild-type S tail samples. **C, D** BLI

assays showing αWD40 interactions of, **C** wild-type and, **D** clientized S tails, the latter being substantially stronger but slower. The concentration of WD40 domains used in BLI assays are shown on the right. **E** PRE plots of Iox/Ired versus residue for Cys1253Ala S tail 21-mer peptide, Cys1253Ala-Thr1273Glu clientized S tail, and a control for intermolecular effects. PRE experiments were performed in duplicate for the Cys1253Ala A tail peptide and intermolecular control, and in triplicate for the Cys1253Ala-Thr1273Glu clientized S tail. The plots show the mean values (bars, points) and error bars are for ±1 standard deviation for $n = 2$ (SARS-CoV-2 wild-type S tail), $n = 3$ (SARS-CoV-2 Cys1253Ala-Thr1273Glu clientized S tail), and $n = 2$ (inter-peptide PRE) biologically independent samples. The position of the MTSL spin label is indicated (orange arrow).

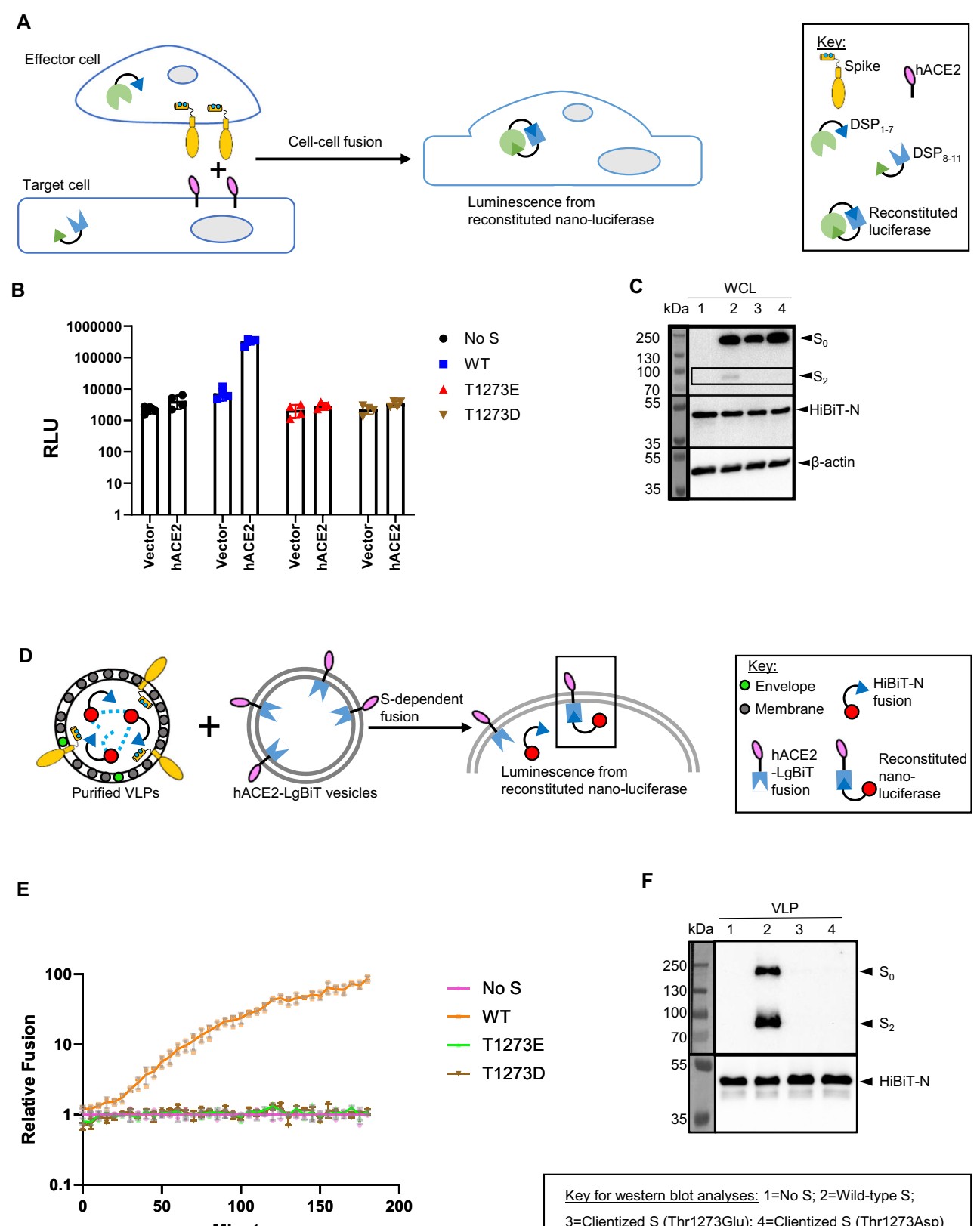

sharp contrast, VLPs from cells expressing clientized Thr1273Asp or Thr1273Glu S proteins failed to generate luminescence and contained few S molecules (Fig. 5E, F). Thus, strong association between clientized S and coatomer interferes with incorporation of S into newly assembling VLPs and compromises VLP-directed membrane fusion.

## The retrograde pathway traffics S protein to the virion assembly site

Finally, to determine whether the anterograde or retrograde pathway is the primary conduit for S protein during SARS-CoV-2 assembly, we utilized a Lys1269Ala/His1271Ala double mutant of S dibasic motif, which is unable to interact with coatomer[10,11]. This retrograde

**Fig. 5 | Clientization inhibits S protein fusogenicity and incorporation in VLPs.**
**A** Schematic of S protein fusogenicity assay. If fusion-competent S proteins traffic to the surface of S-expressing "effector" HEK293T cells (co-express S, M, E, and N-HiBiT), they engage with "target" HEK293T cells expressing the human receptor for SARS-CoV-2 (hACE2) and activate cell-cell fusion. This results in multi-nucleate syncytia, allowing DSP (dual split protein) reconstitution of nano-luciferase from DSP$_{1-7}$ and DSP$_{8-11}$. Reconstitution does not occur if S is fusion incompetent or not trafficked to the PM (plasma membrane). The illustration in panel A was created using Microsoft PowerPoint. **B** Nano-luciferase luminescence for wild-type, and Thr1273Glu or Thr1273Asp clientized S in protein fusogenicity assays. Clientized S shows 10-fold lower nano-luciferase luminescence than wild-type S. Individual data points, mean, and standard error are shown for $n = 4$ biologically independent samples. **C** Western blots of whole cell lysates (WCLs) from effector cells used for protein fusogenicity assays show intact S (S$_0$) from early secretory compartments

and cleaved S$_2$ fragments from PM when wild-type S is expressed. In contrast, no proteolyzed fragments are observed when Thr1273Glu or Thr1273Asp clientized S is expressed. This represents one of two biologically independent samples.
**D** Schematic of SARS-CoV-2 VLP assembly assay. S, M, E, and HiBiT-N plasmids co-transfected in HEK293T cells generate VLPs, which are purified from the extra-cellular medium. VLPs displaying functional S proteins will engage vesicles displaying hACE2 fused to a cytosolic LgBiT fragment, resulting in VLP-vesicle fusion, nano-luciferase reconstitution, and a luminescence signal. The illustration in panel D was created using Microsoft PowerPoint. **E** Luminescence signals from VLP assembly assay. Only wild-type S results in fusogenic activity and enhanced luminescence. Individual data points, mean, and standard error are shown for $n = 3$ biologically independent samples. **F** Western blots of purified VLPs show the abundant presence of wild-type S and the absence of Thr1273Glu or Thr1273Asp clientized S. This represents one of two biologically independent samples.

trafficking-defective S mutant instead undergoes selective antero-grade trafficking. VLPs secreted from cells containing this S mutant resulted in ~75% less membrane fusion than VLPs containing wild-type S (Fig. 6A). Moreover, these retrograde trafficking-defective S molecules were less efficiently incorporated into VLPs (Fig. 6B, C). Interestingly, retrograde trafficking-defective S protein caused 2.3-fold higher fusogenicity of cells in culture than wild-type S protein (Fig. 6D), consistent with its facile escape from the coatomer and consequent accumulation at the PM.

## Discussion

In this study, we used structure-function analyses to determine the molecular function of the non-acidic C-terminal residue of the SARS-CoV-2 S protein, revealing the importance of this lack of mimicry of coatomer interacting residues. Our work revealed the atomic basis of the S-coatomer interaction as well as fundamental details of S protein trafficking and progeny assembly.

In particular, our investigation showed that enhancing mimicry of the client protein extended motif by an acidic residue substitution at Thr1273 substantially increases coatomer binding, thereby localizing Thr1273Glu clientized S protein in coatomer-enriched early secretory compartments. Furthermore, this manipulation provided a means to show that the balance between coatomer-dependent retention and release of S protein is critical for SARS-CoV-2 propagation. If the S-coatomer interaction is enhanced, the assembly of fusion-competent VLPs and S-dependent cell-cell fusion are greatly reduced. Alternatively, if the S-coatomer interaction is abolished, VLP fusogenicity is reduced while cell-cell fusion is increased due to greater anterograde S transport. Hence, an intermediate S-coatomer interaction affinity provides a trade-off between S incorporation into VLPs and cell-cell fusion, allowing both virus transmission within and between host animals. Here, it is important to note that although S incorporation in VLPs is compromised both by substitutions that enhance S-coatomer binding and those that abolish coatomer binding, the underlying mechanism is likely different. Exclusive anterograde trafficking of S in the absence of coatomer-binding potentially reduces M-protein association, which is critical for S incorporation into VLPs. In contrast, enhanced coatomer-binding due to clientization likely prevents S release, thereby affecting S incorporation into virions. It must be noted that clientization does not interfere with the biosynthesis of the S protein, but rather exerts its effect by modifying S localization, fusogenic activity, and VLP incorporation. As such, a single residue at the C-terminus of the S tail is a critical determinant of SARS-CoV-2 propagation and infectivity and explains why acidic C-terminal residues have not been reported in CoV S molecules that display a dibasic motif.

We also elucidated the structural and biophysical basis of the balance between binding and dissociation of the S protein and coatomer. We showed that clientization of the S protein by mutation of Thr1273 to glutamate generates electrostatic complementarity between the acidic side chain of glutamate and a cluster of two basic

residues conserved in αWD40 and β′WD40 domains. The main chain carboxylate of Thr1273 and of Glu1273 interacts with Lys15, the third residue in this basic cluster. Site-directed mutagenesis of this basic cluster showed that an arginine at position 300 in αWD40 determines this enhanced affinity. In contrast, Arg13 and Lys15 residues in αWD40 provide the basis for basal binding to both wild-type and clientized S molecules. Hence, the WD40 basic cluster has evolved distinct and fine-tuned interaction sites that modulate binding affinities for CoV S protein and client proteins. Our analysis thus reveals key atomic-level selectivity principles in coatomer-client interactions. Prior research has focused mainly on two key differences in the WD40 domains, i.e., His31/Tyr139 in αWD40 are replaced by Tyr33/Phe142 in β′WD40 domain, respectively[43,61]. It has been inferred that His31 and Tyr139 provide favorable side-chain interactions for client tail binding that are absent from β′WD40 domain. Furthermore, Tyr33 was suggested to be inhibitory to the binding of client tails to β′WD40 domain wherein the penultimate tail residue is β-branched[61]. Building on these previous data, our present investigation shows that the principles of client-WD40 selectivity are more complex. In particular, the C-terminal acidic residue in client proteins modulates coatomer subunit binding affinity, and hence selectivity. In fact, clientization at the S tail C-terminus substantially overcomes selectivity against β′WD40 domain. Moreover, β′WD40 Tyr33 plays a critical role in stabilizing S tail binding through interactions with the tail Tyr1272 side-chain. Hence, we infer that this selectivity in favor of αWD40 and against β′WD40 domain is not absolute but instead involves relative modulation of interaction affinity for one coatomer subunit versus the other through several interactions at the client-WD40 interface. Furthermore, our data suggest that simultaneous, high avidity engagement of α and β′-COPI subunits likely involves the C-terminal residue in endogenous oligomeric clients such as the ER-resident enzyme UGT[68], which is critical for xenobiotic processing and displays a dibasic motif in its cytosolic tail. Because the dibasic motif is displayed by a variety of proteins such as signaling proteins, enzymes, and growth factors[43], the structural principles of coatomer-client affinity modulation that we have elucidated have broad implications for secretory homeostasis.

Our analyses of SARS-CoV-2 VLP assembly revealed that incorporation of retrogradely trafficked S protein is favored over newly synthesized protein delivered by anterograde trafficking from the ER. Golgi-associated S protein may be identified for retrograde trafficking by its association with Golgi-resident CoV M proteins[3,12], consistent with its dependence on cis-Golgi-located M protein for incorporation into virions[6,12]. However, because the CoV M protein lacks a dibasic retrieval motif, our data suggest that the S protein acts as a chaperone to deliver M to ERGIC. Once delivered, M protein may direct S into budding virions. Our investigation also raises intriguing questions about the balance between anterograde and retrograde trafficking of S at various stages of infection. Presumably, newly synthesized S is at a low concentration during the early stages of infection and therefore efficiently retrieved by the coatomer for incorporation into virions.

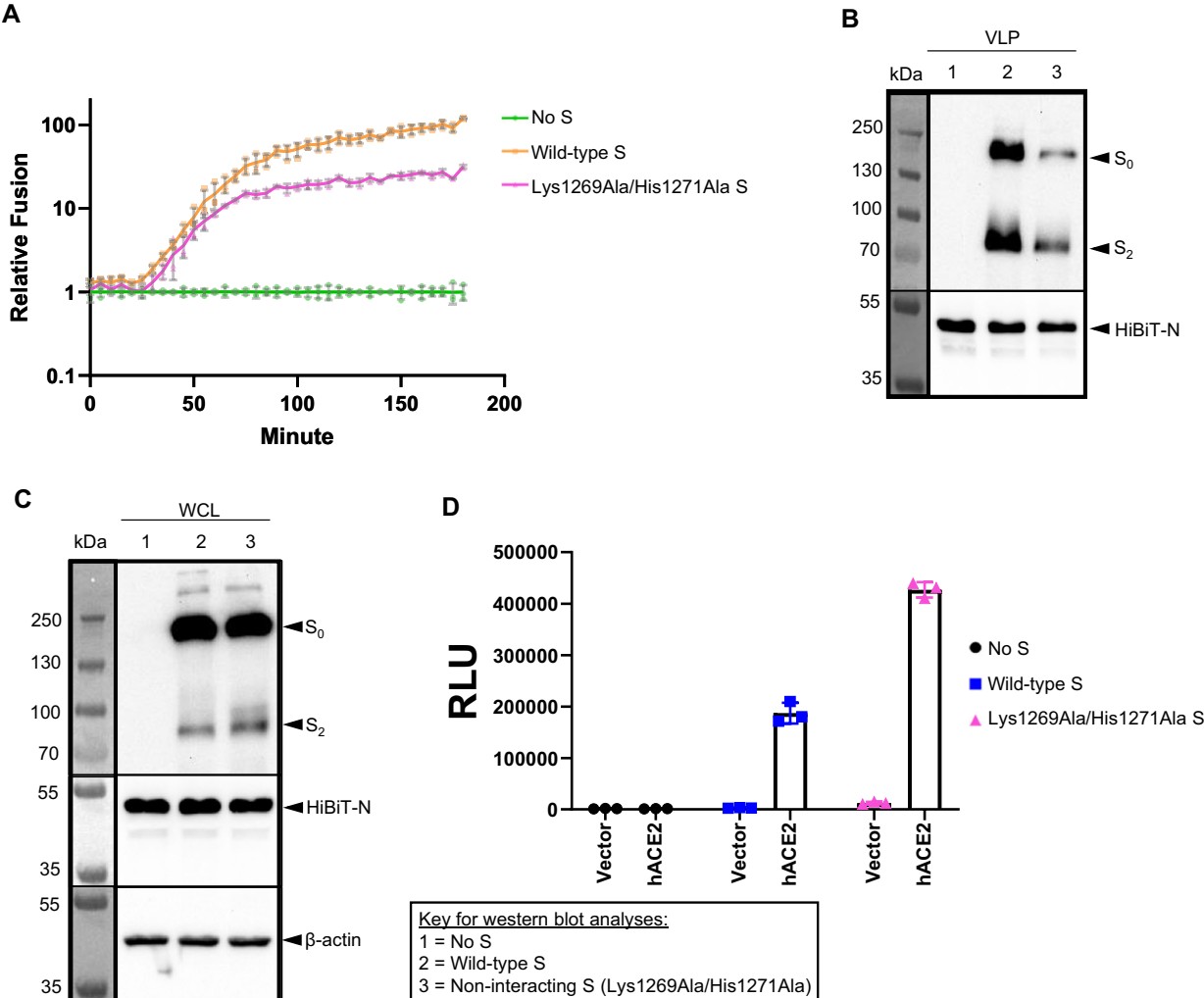

**Fig. 6 | Coatomer binding facilitates incorporation of S into VLPs and inhibits cell-cell fusogenicity. A** Luminescence signals from VLP assembly assay. An S protein double mutant (Lys1269Ala/His1271Ala) that does not interact with coatomer shows a substantially smaller increase in luminescence than wild-type indicating poor incorporation into VLPs. Individual data points, mean, and standard error are shown for $n$ = 3 biologically independent samples. **B** Western blots of purified VLPs confirm poor incorporation of the non-interacting S protein. **C** Western blots of WCLs (whole cell lysates) show no expression differences between the wild-type and non-interacting S protein. This represents one of two biologically independent samples. **D** Nano-luciferase luminescence for non-interacting S protein is 2.3-fold higher than wild-type. Individual data points, mean, and standard error are shown for $n$ = 3 biologically independent samples.

However, higher concentrations of S in later stages are likely to saturate the coatomer, resulting in the export of S to the PM for cell-cell fusion. This leads to an interesting hypothesis, i.e., early in infection, the coatomer sequesters the S protein near ERGIC to provide a window for the assembly of infectious progeny virions, prior to S export to PM at a later stage for cell-cell fusion and virion transmission. Both the hypotheses related to this time delay and the evolutionary specialization of M and S proteins require further investigation.

Although retrograde and anterograde trafficking of the S protein are critical for SARS-CoV-2 propagation, the mechanism underlying trafficking bifurcation at the cis-Golgi is not well understood. Structural insights from our NMR studies advance the understanding of this intriguing mechanism. S molecules in the cis-Golgi with an inaccessible extended motif will be incompetent for coatomer binding and continue anterograde export to the PM. In contrast, an accessible extended motif will permit S retrieval to the virion assembly site in ERGIC. The NMR data presented here suggest a turn-like structure of the wild-type S tail, which can potentially present conformational constraints on coatomer interactions. Furthermore, the inherent conformational flexibility in the S tail likely enables exposure of the extended motif for coatomer binding. Such an equilibrium in S tail states might be modulated by factors such as oligomerization, palmitoylation, proximity to the Golgi membrane, association with cytosolic proteins, and mutations outside the extended motif[11,69,70]. For example, mutagenesis of the SARS-CoV-2 S tail has identified residues upstream of the extended motif (Ser1261-Glu1262) that are critical for coatomer interactions[11]. Our structural analysis maps these two residues to a turn in the S tail, suggesting that the Ser1261Ala/Glu1262Ala double mutation modifies the conformational accessibility of the extended motif and reduces coatomer binding. In contrast, the Thr1273Ala substitution likely modifies the S tail conformational accessibility for coatomer binding, which results in its enhanced retention in early secretory compartments[11]. On a local level, our data show a function of an acidic residue at the S protein C-terminus in modulating dibasic motif accessibility. Given that several client proteins display acidic residues at the C-terminal position, our data are suggestive of a broad modulatory function of this residue in secretory trafficking. Overall, our analysis reveals that indirect, long-range conformational dynamics as well as local interactions determine S-coatomer binding and the balance of anterograde-retrograde trafficking.

Bifurcation of S trafficking has implications for vaccinology. Our data suggest that the C-terminal residue of S protein encoded by

mRNA or adenoviral constructs[1] could modulate S export and its antigenic display on the cell surface. Such C-terminus optimization will be essential for S export and a robust immune response to genetic vaccines against the wide diversity of C-terminal residues in S proteins encoded by CoVs of human concern, including the lethal MERS-CoV and human CoV OC43 associated with common cold-like symptoms[43,71]. Furthermore, our research raises important questions about the modulation of endogenous protein trafficking upon transient expression of S-based genetic vaccines, as disruption of coatomer function has serious clinical outcomes. For instance, disruption of client retrograde retrieval due to mutations in COPA, the human ortholog of α-COPI, underlies the autoimmune COPA syndrome, which causes coughing, shortness of breath, lung inflammation and scarring, pulmonary hemorrhaging, and systemic autoimmune reactions such as arthritis[46]. This emphasizes an urgent need to investigate secretory disruptions due to coatomer hijacking by SARS-CoV-2 or by transient expression of S-based genetic vaccines.

Our investigation addresses the fundamental basis of a popular approach in coatomer-client structural biology, in using β'WD40 as a substitute for αWD40 for client tail peptide co-crystallization. We have recently reported that there is a local structural similarity between an αWD40 loop and the N-terminal Lys residue in the client tail dibasic motif[72]. As such, the client tail peptides are often displaced by this αWD40 loop. Hence, although the αWD40 is favored by several clients in binding assays, it is often not suitable for co-crystallization with client peptides. Instead, β'WD40 domain has been used as a substitute for αWD40 domain in co-crystallization even though this domain shows minimal binding to client tail peptides in biophysical approaches such as BLI and ITC. Our NMR analysis, which is substantially more suitable for assaying weak interactions, clearly shows evidence of S tail-β'WD40 interactions that are weaker than those with αWD40 domain. It is likely that high concentrations in crystallization push the equilibrium towards peptide-β'WD40 binding, as suggested previously[61]. Hence, this analysis provides a biophysical basis for client tail co-crystallization with β'WD40 domain.

In summary, we have shown that the bipartite organization of the S protein extended motif balances two opposing forces: coatomer hijacking to traffic S to the ERGIC and S-coatomer dissociation at this assembly site. Together with histone mimicry by SARS-CoV-2 to modify the host epigenome and anti-viral response[73], our investigation establishes molecular mimicry as a key mechanism employed by SARS-CoV-2 to achieve infection and progeny assembly.

## Methods

### Data reporting
No statistical methods were used to predetermine the sample size. The experiments were not randomized, and the investigators were not blinded to allocation during experiments and outcome assessment.

### Plasmids
All plasmids in this investigation will be publicly available through Addgene.

### Antibodies
The following antibodies were used in this investigation: SARS-CoV/SARS-CoV-2 S monoclonal antibody 1A9 (Cat. No. GTX632604; GeneTex, USA; 1:1000 dilution); goat anti-mouse IgG (H + L) highly cross-adsorbed secondary antibody, Alexa Fluor™ Plus 488 (Cat. No. A-11001; Invitrogen, USA; 1:700 dilution); goat anti-rabbit IgG (H + L) highly cross-adsorbed secondary antibody, Alexa Fluor™ 555 (Cat. No. A32732; Invitrogen, USA; 1:700 dilution); SARS-CoV-2 (2019-nCoV) S antibody, rabbit polyclonal antibody, antigen affinity purified (Cat. No. 40591-T62; Sino Biological, USA; 1:1000 dilution); SARS-CoV/SARS-CoV-2 S monoclonal antibody 1A9 (Cat. No. MA5-35946; Thermo Fisher Scientific, USA; 1:1000 dilution); goat anti-human IgG-HRP (Cat. No. sc-2907; Santa Cruz Biotechnology Inc, USA; 1:5000 dilution); ACE2 recombinant rabbit monoclonal antibody (SN0754) (Cat. No. MA5-32307; Thermo Fisher Scientific, USA; 1:1000 dilution); mouse anti-β-actin-peroxide monoclonal antibody (Cat. No. A3854; Sigma-Aldrich, USA; 1:10,000 dilution); rabbit anti-β-COPI polyclonal antibody (Cat. No. ab2899; Abcam, USA; 1:1000 dilution); rabbit anti-giantin polyclonal antibody (Cat. No. 924302; BioLegend, USA; 1:1000 dilution).

### Eukaryotic cell lines
The eukaryotic cell lines used in this manuscript are as follows: Expi293F (Cat. No. A14527, Thermo Fisher Scientific, USA), HEK293T (Cat. No. CRL-3216, ATCC), Vero-E6 (Cat. No. CRL-1586, ATCC), and HeLa (Cat. No. CCL-1, ATCC).

### Immunofluorescence microscopy
Constructs encoding either wild-type full-length S protein or the clientized Thr1273Glu S mutant, were transfected using jetOPTIMUS transfection reagent (Polyplus, France) into HeLa cells on sterile glass coverslips. Then, these cells were fixed the next day with 4% formaldehyde (Sigma-Aldrich, USA) for 10 min. Permeabilization and blocking were performed for 1 hour with PBS (with 0.4% (v/v) Triton X-100) and 2% immunoglobulin G-free BSA (Jackson ImmunoResearch, USA). The anti-S 1A9 mouse monoclonal antibody in PBS (with 0.1% Triton X-100% and 0.5% BSA) was used to probe the cells for S protein localization. These cells were then mounted in ProLong® Glass antifade mounting medium (Life Technologies, USA), followed by treatment with fluorophore-conjugated goat anti-mouse IgG (H + L) Highly Cross-Adsorbed Secondary Antibody, Alexa Fluor™ Plus 488 and goat anti-rabbit IgG (H + L) Highly Cross-Adsorbed Secondary Antibody, Alexa Fluor™ 555 antibody and subsequent washing. An LSM880 confocal microscope (Carl Zeiss Inc, Germany) was used to acquire images, which were then analyzed by ImageJ software[74].

### Preparation of mouse liver cytosol (MLC)
Fresh C57Blk6 mouse livers were kindly provided by Dr. Stephen Oh (Washington University School of Medicine, St. Louis MO). Liver tissues (10 g) were chopped into small pieces on a petri plate on ice and homogenized with 20 ml homogenization buffer A (25 mM HEPES-KOH, pH 7.4, 125 mM potassium acetate, 2.5 mM magnesium acetate, 1 mM DTT) using a Dounce glass homogenizer (50 strokes). The homogenized extract was first centrifuged at 800 x $g$ for 10 mins at 4 °C to remove nuclei, then centrifuged again at 10,000 x $g$ for 10 mins at 4 °C to remove mitochondria. The resulting supernatant was then subject to ultracentrifugation at 100,000 x $g$ for 45 minutes at 4 °C to separate membranes from cytosol. The clarified cytosolic fraction was then carefully removed and stored in 1 ml aliquots at −80 °C until further use in pull-down assays.

### GST pull-down assays
For GST pull-down assay, a tube of the 1 ml mouse liver cytosol was thawed on ice, centrifuged at 20,000 x $g$ for 20 minutes at 4 °C to remove any precipitated material, and the clarified cytosol was then diluted to a total protein concentration of approximately 20 mg/ml with cold homogenization buffer B (homogenization buffer A containing TX-100 at a final concentration of 0.1%). For binding assays, 100 µg of each purified fusion protein was first immobilized on 50 µl of packed glutathione-agarose beads at room temperature for 1 hour, the beads were washed once with cold homogenization buffer B, then 300 µl of the diluted cytosol was added and the beads tumbled for 2 h at 4 °C. The beads were washed 3 times with cold homogenization buffer B to remove non-specifically bound proteins, by gentle centrifugation at 2000 x $g$ for 10 seconds at 4 °C in a table-top centrifuge The bead pellet was resuspended in SDS-PAGE sample buffer and heated at 100 °C for 10 mins before analysis by SDS-PAGE.

## Western blot and antibodies

In Fig. 1, the anti-S 1A9 mouse monoclonal antibody (catalog # GTX632604, GeneTex, USA) was used in the western blot analysis. In Figs. 5 and 6, samples in SDS solubilizer (0.0625 M Tris·HCl (pH 6.8), 10% glycerol, 0.01% bromophenol blue, 2% (wt/vol) SDS, +/- 2% 2-mercaptoethanol) were heated at 95 °C for 5 min, electrophoresed through 8% or 10% (wt/vol) polyacrylamide-SDS gels, transferred to nitrocellulose membranes (Bio-Rad, USA), and incubated with SARS-CoV-2 (2019-nCoV) S antibody rabbit PAb antigen affinity purified, mouse monoclonal anti-SARS-S2 conjugated to HRP, goat anti-human IgG-HRP, ACE2 recombinant rabbit monoclonal antibody, or purified LgBiT-substrate cocktail. After incubation with appropriate HRP-tagged secondary antibodies and chemiluminescent substrate (Thermo Fisher, USA), the blots were imaged and processed with a ProteinSimple FlourChem E (Bio-Techne, USA).

## Coatomer WD40 expression and purification

Recombinant expression and purification of *S. pombe* αWD40 and *S. cerevisiae* β'WD40 domains (both wild type and mutants) were carried out as described[43,72,75]. Briefly, αWD40 with a C-terminal strep-tag was expressed and purified from Expi293 cells using polyethyleneimine (PEI). The clarified cell lysate was subjected to affinity chromatography followed by size-exclusion chromatography (SEC) on a preparative grade Superdex-75 column.

β'WD40 domain with an N-terminal strep-Hisx6-SUMO tag was expressed in BL21(DE3)pLysS cells by IPTG induction at 18 °C overnight. After affinity chromatography, β'WD40 was further subjected to digestion with the ULP1 protease (Kerafast, USA) for removal of the SUMO tag. Further purification was performed Ni-NTA chromatography and SEC. Plasmids for all WD40 mutants (αWD40 Arg13Ala, Lys15Ala, Arg300Ala, and β'WD40 Tyr33Ala, Lys17Ala) were generated either using the Q5 Site-Directed Mutagenesis Kit (New England Biolabs, USA) or were synthesized by TOPGENE Technologies (Canada) and were purified as described above.

## Biolayer interferometry (BLI)

BLI assays with biotinylated S peptides from Biomatik (USA) and purified coatomer WD40 domains were carried out as described[43]. Briefly, the purified WD40 domain was provided as the analyte with a buffer composition of 20 mM Tris·HCl (pH 7.5) or 50 mM MES-NaOH (pH 6.5), 150 mM NaCl, 5 mM DTT, 10% glycerol, 0.2 mg/ml bovine serum albumin (BSA), and 0.002% Tween 20. S peptides were immobilized on streptavidin (SA) biosensors (FortéBio, USA). Measurements were performed on the Octet RED96 system (FortéBio, USA). Data acquisition and data analysis were performed using Data Acquisition 11.1 suite and FortéBio Data Analysis 11.1 software suite respectively. All experiments were performed at 25 °C and 1000 rpm shake speed. Appropriate sensor and loading controls were applied. All BLI sensorgrams reported in this investigation were within statistical limits, i.e., $R^2 > 0.98$ and $\chi^2 < 3$.

## X-ray crystallography

Purified mutant αWD40 domains were concentrated to ~2 mg/ml. Crystal trays were set-up using the hanging drop vapor diffusion method with αWD40 domains mixed with reservoir buffer in 1:1 v/v ratio. Crystals grew at room temperature within 24 hours. Crystals of αWD40 Arg13Ala grew in 0.14 M sodium citrate tribasic dehydrate, 16% PEG3350. Crystals of αWD40 Lys15Ala grew in 0.14 M sodium citrate tribasic dihydrate, 18% PEG3350. Crystals of αWD40 Arg300Ala grew in 0.2 M potassium sodium tartrate tetrahydrate, 18%PEG3350.

The purified β'WD40 domain was concentrated to 20-25 mg/ml. Co-crystals of β'WD40 domain with wild-type and clientizing S hepta-peptides were set-up with a WD40:peptide molar ratio of 1:2. Co-crystals grew in 2-4 days at room temperature. Wild-type β'WD40 with wild-type S hepta-peptide grew in 0.1 M MES pH 6.2, 19% PEG 4000.

Wild-type β'COPI WD40 with clientizing S hepta-peptide grew in 0.1 M MES pH 6.0, 15% PEG 20000. Co-crystals of β'WD40-Tyr33Ala mutant with clientizing S hepta-peptide grew in 0.1 M MES pH 6.2, 17% PEG 20000. Crystals of β'WD40-Lys17Ala mutant grew in 0.1 M MES pH 6.5, 19% PEG 20000.

X-ray diffraction data collection was carried out at Brookhaven National Laboratory, National Synchrotron Light Source II (NSLS II) beamline 17-ID-1 AMX[76,77]. Diffraction data was indexed, integrated, and scaled in XDS[78] as part of the beamline data acquisition and processing pipeline. Data processing statistics are in Supplementary Tables S3 and S4. Molecular replacement was performed in Phenix using PDB ID's 7S22 for αWD40 mutants and 4J79 for β'WD40 in complex with S hepta-peptides[43,61,79]. Refinement and model building were performed using Phenix.refine[79] and Coot[80]. All figures were generated using Pymol and Coot.

## Preparation of isotope-labeled S tail peptides

A gene fragment corresponding to the wild-type SARS-CoV-2 S C-terminal tail was cloned into an eXact tag pH720 vector system[81]. Isotope-labeled ($^{15}$N, $^{13}$C/$^{15}$N) peptide was prepared by transforming the construct into *E. coli* BL21(DE3) cells, followed by growth in M9 minimal media with either $^{15}$N-ammonium chloride or $^{13}$C-glucose/$^{15}$N-ammonium chloride at 37 °C until an OD600 ~ 0.6-0.9 was reached. Protein expression was induced with 1 mM IPTG for 18 hours at 25 °C. Cells were centrifuged, re-suspended, and lysed using sonication. The cleared lysate was purified on an immobilized subtilisin column (Potomac Affinity Proteins, USA), using a method similar to that described previously[82]. Fractions containing at least 95% pure peptide, as determined by MALDI, were pooled and concentrated for further analysis. The Q5 site-directed mutagenesis kit (New England Biolabs, USA) was used for the preparation of mutants.

## NMR spectroscopy

NMR spectra were acquired on a Bruker Avance III 600 MHz spectrometer fitted with a Z-gradient $^{1}$H/$^{13}$C/$^{15}$N-cryoprobe. Backbone resonance assignments for the 21-residue SARS-CoV-2 S C-terminal tail peptide were made using heteronuclear triple resonance experiments as follows: HNCACB, CBCA(CO)NH, HNCO, HN(CA)CO, and (H)N(CA)NNH. Sample conditions were 300 μM peptide, 100 mM KPi, 1 mM DTT, pH 7.0, 25 °C. Secondary shifts were calculated from experimental and sequence-corrected random coil chemical shifts[83]. Spectra were processed with NMRPipe[84] and analyzed with Sparky[85].

Chemical shift mapping experiments were performed by comparing two dimensional $^{1}$H-$^{15}$N HSQC spectra of $^{15}$N-labeled SARS-CoV-2 S tail in its unbound and WD40-bound states. Bound states were prepared by adding unlabeled αWD40 or β'WD40 to $^{15}$N-labeled peptide with a WD40/peptide molar ratio of ~1.2:1. Chemical shift perturbations were calculated from the equation, $\Delta\delta_{total} = [(W_H \, \Delta\delta_H)^2 + (W_N \, \Delta\delta_N)^2]^{1/2}$, where WH = 1 and Wn = 0.2[82]. To allow for more direct comparison with the BLI results, sample conditions for shift mapping were 20 mM Tris, 150 mM sodium chloride, 5 mM DTT, pH 7.5, 25 °C. This produced only minor changes in the HSQC spectrum of the unbound state so that assignments were readily transferred from above.

PRE measurements were made using $^{15}$N-labeled Cys1253Ala and Cys1253Ala/Thr1273Glu mutants of the S tail peptide that were reacted with 10 molar equivalents of MTSL (Santa Cruz Biotechnology, USA) for 1-2 hours at room temperature. Reactions were checked for completion by MALDI and excess MTSL was removed by dialysis. Two-dimensional $^{1}$H-$^{15}$N HSQC spectra were acquired on 100 μM MTSL-derived tail peptides before and after reduction with 20 molar equivalents of sodium ascorbate. Peak intensities for the oxidized and reduced states, $I_{ox}$ and $I_{red}$, were determined with SPARKY. To control for intermolecular effects, a natural abundance sample of the MTSL-peptide was mixed in a 1:1 ratio with a $^{15}$N-labeled Cys1253Ser/

 

Cys1254Ser mutant S tail peptide, and peak intensities were measured for the oxidized and reduced states as above.

## VLPs

HEK293T (obtained from Dr. Ed Campbell, Loyola University Chicago) and Vero-E6 (ATCC CRL-1586) cells were maintained in DMEM-10% FBS (Dulbecco's Modified Eagle Media, DMEM) containing 10 mM HEPES, 100 nM sodium pyruvate, 0.1 mM non-essential amino acids, 100 U/ml penicillin G, and 100 μg/ml streptomycin, and supplemented with 10% fetal bovine serum (FBS, Atlanta Biologicals, USA). These cell lines were cultured in a 5% CO2 incubator at 37 °C. Full-length SARS-CoV S (GenBank: AY278741.1) and SARS-CoV-2 S, E, M, and N (GenBank: NC_045512.2) genes were synthesized by Genscript Inc (USA) as human codon-optimized cDNAs, and inserted into pcDNA3.1 expression vectors. HiBiT-N was constructed by fusing HiBiT peptide (VSGWRLFKKIS) coding sequences with linker (GSSGGSSG) to the 5′ end of the N gene, as described[66,67,86]. The pCMV-LgBiT expression plasmid was purchased from Promega (USA). pcDNA3.1-hACE2-C9 was obtained from Dr. Michael Farzan, Scripps Florida. pcDNA3.1-hACE2-LgBiT was constructed by fusing the coding sequence of LgBiT to the 3′ end of hACE2 gene. HiBiT-N tagged VLPs were produced as described previously[66,67,86]. Briefly, equimolar amounts of full-length CoV S, E, M and HiBiT-N encoding plasmids (total 10 ug) were LipoD (catalog # SL100668, SignaGen, USA)-transfected into 10^7 HEK293T cells. To produce S-less "No S" VLPs, the S expression plasmids were replaced with empty vector plasmids. At 6 h post-transfection, cells were replenished with fresh DMEM-10% FBS. HiBiT-N VLPs were collected in FBS-free DMEM from 24 to 48 h post-transfection. FBS-free DMEM containing HiBiT-N VLPs were clarified by centrifugation (300 x g, 4 °C, 10 min; 3000 x g, 4 °C, 10 min). To obtain purified VLPs, clarified VLP-containing FBS-free DMEM was concentrated 100-fold by ultracentrifugation (SW28, 7500 rpm, 4 C, 24 h) through 20% (w/w) sucrose[87]. VLPs were quantified after detergent lysis by adding LgBiT and measuring complemented Nluc in a luminometer. For downstream experiments, VLP inputs were normalized based on their Nluc activity upon LgBiT complementation. VLP samples were stored at −80 °C.

## Cell-free fusion assay

hACE2-LgBiT extracellular vesicles (EVs) were obtained as described previously[66,86]. Briefly, HEK293T target cells were LipoD-transfected with pcDNA3.1-hACE2-LgBiT. At 6 h post-transfection, transfection media were removed, rinsed, and replaced with FBS-free DMEM. Media were collected at 48 hours post-transfection, clarified (300 x g, 4 °C, 10 min; 3000 x g, 4 °C, 10 min), and concentrated 100-fold by ultrafiltration (Amicon, 100 kDa). EVs were then purified using size-exclusion chromatography (qEV Original, Izon Science, New Zealand) using PBS pH 7.4 as eluant. Peak EV fractions were identified by adding HiBiT-containing detergent and subsequent nano-luciferase (Nluc) measurement by luminometry. EVs were stored at 4 °C. Cell-free fusion assays were performed as described previously[66,86]. Briefly, at 4 °C, equal volumes of HiBiT-N VLPs and hACE2-LgBiT EVs were mixed with nanoluc substrate (catalog # N2420, Promega, USA) and trypsin (Sigma-Aldrich, USA; 10 ng/μl or as indicated) in 384-well multiwell plates. Sample plates were then loaded into a Glomax luminometer maintained at 37 °C. Nluc accumulations were recorded over time. VLP-EV cell-free fusions were quantified as the fold increase of Nluc signal from S-bearing VLPs over the signal from S-less (no S) VLP background control.

## Cell-cell fusion assay

VLP-producer cells and target cells were prepared as described previously[66]. Briefly, VLP-producer HEK293T cells were co-transfected with pDSP1-7 and expression plasmids for SARS-CoV-2 S, E, M, and N-HiBiT. Control effector cells received empty vector plasmid instead of S plasmid. Target cells (HEK293T) were co-transfected with pDSP8-11 and empty vector or hACE2-expressing plasmids. At 6 hours post-transfection, VLP-producer and target cells were suspended and mixed into white-walled 96-well plates. 18 hours later, live-cell renilla luciferase (Rluc) substrate (EnduRen, Promega, USA) was added. After 2 hours, Rluc levels were quantified using a Veritas microplate luminometer.

## Bottom-up proteomics

Samples were spiked with 10x lysis buffer (10 μL, 120 mM sodium lauroyl sarcosine, 5% sodium deoxycholate, 500 mM triethylammonium bicarbonate (TEAB), Halt™ Protease and Phosphatase Inhibitor Cocktail (Thermo Fisher Scientific, USA). The samples were treated with tris (2-carboxyethyl) phosphine (10 μL, 55 mM in 50 mM TEAB, 30 min, 60 °C) followed by treatment with chloroacetamide (10 μL, 120 mM in 50 mM TEAB, 30 min, 25 °C in the dark). They were then diluted 5-fold with aqueous 50 mM TEAB and incubated overnight with Sequencing Grade Modified Trypsin (1 μg in 10 uL of 50 mM TEAB; Promega, USA). Following this an equal volume of ethyl acetate/trifluoroacetic acid (TFA, 100/1, v/v) was added and after vigorous mixing (5 min) and centrifugation (13,000 x g, 5 min), the supernatants were discarded, and the lower phases were dried in a centrifugal vacuum concentrator. The samples were then desalted using a modified version of Rappsilber's protocol[88] in which the dried samples were reconstituted in acetonitrile/water/TFA (solvent A, 100 μL, 2/98/0.1, v/v/v) and then loaded onto a small portion of a C18-silica disk (3 M, USA) placed in a 200 μL pipette tip. Prior to sample loading the C18 disk was prepared by sequential treatment with methanol (20 μL), acetonitrile/water/TFA (solvent B, 20 μL, 80/20/0.1, v/v/v) and finally with solvent A (20 μL). After loading the sample, the disc was washed with solvent A (20 μL, eluent discarded) and eluted with solvent B (40 μL). The collected eluent was dried in a centrifugal vacuum concentrator and reconstituted in water/acetonitrile/FA (solvent E, 10 μL, 98/2/0.1, v/v/v). Aliquots (5 μL) were injected onto a reverse phase nanobore HPLC column (AcuTech Scientific Inc, USA; C18, 1.8 μm particle size, 360 μm x 20 cm, 150 μm ID), equilibrated in solvent E and eluted (500 nL/min) with an increasing concentration of solvent F (acetonitrile/water/FA, 98/2/0.1, v/v/v: min/% F; 0/0, 5/3, 18/7, 74/12, 144/24, 153/27, 162/40, 164/80, 174/80, 176/0, 180/0) using an EASY-nLC II (Thermo Fisher Scientific, USA). The effluent from the column was directed to a nanospray ionization source connected to a hybrid quadrupole-Orbitrap mass spectrometer (Q Exactive Plus, Thermo Fisher Scientific, USA) acquiring mass spectra in a data-dependent mode alternating between a full scan (m/z 350-1700, automated gain control (AGC) target 3 ×106, 50 ms maximum injection time, FWHM resolution 70,000 at m/z 200) and up to 15 MS/MS scans (quadrupole isolation of charge states 2-7, isolation window 0.7 m/z) with previously optimized fragmentation conditions (normalized collision energy of 32, dynamic exclusion of 30 s, AGC target 1 × 105, 100 ms maximum injection time, FWHM resolution 35,000 at m/z 200). The raw data was analyzed in Proteome Discoverer (Version 2.4; Thermo Fisher Scientific, USA), which provided measurements of abundance for the identified peptides. Tryptic peptides containing amino acid sequences unique to individual proteins were used to identify coatomer subunits α, β, β′, δ, ε, γ-1 and ζ-1 in each sample.

## Reporting summary

Further information on research design is available in the Nature Portfolio Reporting Summary linked to this article.

## Data availability

Source data are published as an accompanying file. All plasmids will be available through Addgene. All crystal structures have been deposited in the Protein Data Bank (PDB) with accession IDs: 8ENS (β′WD40 with wild-type S tail heptapeptide), 8ENW (β′WD40 with Thr1273Glu

clientized S tail heptapeptide), 8ENX (β'WD40-Tyr33Ala with Thr1273Glu clientized S tail heptapeptide), 8ENY (αWD40-Arg13Ala), 8ENZ (αWD40-Lys15Ala), 8EO0 (αWD40-Arg300Ala), and 8SZX (β'WD40-Lys17Ala). NMR assignments for the wild-type and mutant S tails have been deposited in the BioMagResBank (BMRB) with accession codes 51663 (doi:10.13018/BMR51663) and 52007 (doi:10.13018/BMR52007), respectively. Source data are provided with this paper.

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

## Acknowledgements

SSH acknowledges support from the University of Maryland School of Medicine (UOMSOM), UOMSOM Center for Biomolecular Therapeutics, Maryland Department of Health's Cigarette Restitution Fund Program (CH-649-CRF), and the University of Maryland Marlene and Stewart Greenebaum Comprehensive Cancer Center (National Cancer Institute - Cancer Center Support Grant (CCSG) - P30CA134274). SSH and BGP acknowledge support from the University of Maryland MPower Award. SSH and JO acknowledge support from the MPower COVID-19 Response Fund Award. SSH, JO, and BD acknowledge support from NIH R01GM150187. BD and BJ acknowledge support from NIH R01 CA008759. TG and EQ acknowledge support from NIH P01 AI060699. This research used resources at beamline 17-ID-1 AMX of the National Synchrotron Light Source II, a U.S. DOE Office of Science User Facility operated for the DOE Office of Science by Brookhaven National Laboratory under contract no. DE-SC0012704. The NMR facility at the University of Maryland Institute for Bioscience and Biotechnology Research is supported by the University of Maryland, the National Institute of Standards and Technology, and a grant from the W. M. Keck Foundation. We thank Dr. Jean Jakoncic and Dr. Dale Kreitler (AMX beamline), Dr. Oleksandr Galkin (Sartorius Corporation), and Prof. John E. Johnson (The Scripps Research Institute) for advice.

## Author contributions

D.D. designed, performed, analyzed, and interpreted BLI and crystallographic experiments, and wrote the paper; E.Q., designed, performed, analyzed, and interpreted VLP and fusion experiments, and wrote the paper; Y.H., designed, performed, analyzed, and interpreted NMR experiments; Y.C., designed, performed, analyzed, and interpreted NMR experiments; B.J., designed, performed, analyzed, and interpreted S cleavage experiments; W.C., performed and analyzed mass spectrometry experiments; S.S., analyzed and interpreted BLI and crystallographic experiments; L.G., analyzed and interpreted BLI and crystallographic experiments; N.J.S., analyzed and interpreted BLI and crystallographic experiments; B.G.P., analyzed and interpreted structural data, and wrote the paper; J.P.W., designed, performed, analyzed, and interpreted mass spectrometry experiments, and wrote the paper; B.D., designed, performed, analyzed, and interpreted coatomer pull-down, S cleavage, and S immunofluorescence experiments, and wrote the paper; J.O., designed, performed, analyzed, and interpreted NMR experiments, and wrote the paper; T.G., designed, performed, analyzed, and interpreted VLP and fusion experiments, and wrote the paper; S.S.H. conceived and supervised the project, designed and directed the research, analyzed and interpreted data, and wrote the paper.

## Competing interests

The authors declare no competing interests.
