## [Peer Review File · Nature Communications]

A single C-terminal residue controls SARS-CoV-2 spike trafficking and incorporation into VLPsREVIEWER COMMENTS

Reviewer #1 (Remarks to the Author):

This manuscript by Hasan et al focuses on essentially the final two residues of the much-studied Coatomer-binding motif of SARS CoV2 S protein. Whilst well written and presented, and appears technically sound and exhaustive, it suffers greatly in its scope, which is narrow and of little interest to a general field as it largely reiterates what has been presented in many other studies. It doesn't really move the field along given all that has been published on it.

There are several issues, which I think need addressing regardless of the outcome of review.

No explanation is really given for why S binds alpha not beta' when its signal is closest to a beta' sequence. There is much discussion later on as to why many structure and modelling based experiments don't appear to work in respect to this and the authors predictions but no firm conclusions

Repeatedly throughout the authors state wt S does not bind beta' and no KD is given and yet it most clearly does bind or they could not have solved its structure – or is all this work based on an artefact? No explanation is given.

Indeed on line 151 it says a YtoA mutation structure was solved with S and three lines later it says YtoA abolishes binding to peptide – which or what is it?

Why use such a short peptide for NMR PRE as this is unrepresentative of in vivo – use the whole cytoplasmic domain to make experiments valid.

Overall, I found the structural work of little general interest, focussing as it does on essentially one or two residues in a highly studied motif, although the cellular work between lines 78 and 100 was interesting, although again I failed to see the quite point of its importance.

Reviewer #2 (Remarks to the Author):

The assembly mechanism of coronavirus assembly is still poorly characterized. The trafficking of the spike protein is dependent on its C-terminal tail. Retrograde trafficking of protein is mediated by the coatomer that recognizes dibasic residues at the C-terminal extremity of proteins (K-x-K-x-xCT or K-K-x-xCt) with the last residue being frequently acidic according to the authors. The spike protein contains a non canonical K-x-H motif that mediates its retrograde trafficking without any acidic residue at its C-terminus.

By analyzing sequences, the authors did not find any mutation of the last residue (T1273) into acidic residue that would increase the mimicry of the spike motif. The author mutated the Thr1273 into Glu and analyzed the effects of the mutation. They show an increase binding of the coatomer and an increased intracellular retention of the protein compared to the wild-type spike. The coatomer is composed of different subunits, among them the β -propellor WD40 domains of the α or β' subunit binds the retrieval signal on proteins, the spike protein has been shown to preferentially binds the α WD40 domains. By using cristal structures and BLI assays using heptapeptide, the authors identified basic residue of the WD40 binding sites that are important for the increased binding of the spike motif bearing a T1273G mutations. Moreover, they show that this mutation allows the binding of the β' WD40 domains, meaning that this last residue is important for the spike-coatomer selectivity. They also characterized more precisely the interactions of S-WD40 domains and the importance of these interactions for WD40 domains selectivity. Then the authors analyzed the effect of the mutation in the context of the full-length C-terminal domain. By using NMR, they found that the C-tail of the wild-type protein may have a more compact conformation. They also confirmed in BLI assay the stronger binding of the C-tail harboring the T1273G mutation to the WD40 domain of the coatomer. Finally, they analyzed the effect of the mutation in cell-cell fusion assay and virus-like particle (VLP) production. The mutation T1273G decreased cell-cell fusion and incorporation of the spike protein in VLP.

These results are originals and provide informations about the selectivity of the spike protein for the WD40 domain of the coatomer and more generally in term of cell biology and protein sorting. Moreover, they show that this non-optimal binding motif is important for the viral cycle and virus propagation. Indeed, optimization of the binding prevents spike incorporation into VLP and cell-cell fusion.

Minor points:

- statistical analysis are missing
- Concerning the figure 5, for the cell-cell fusion assay, without carefull reading of the material and methods, it is not clear that effector cells express all the structural proteins (M, E and HiBiT-N) and the conclusion line 230-231 does not make sense at first. This information should be added in the text or the legend of the figure.
- in the discussion line 261: the authors stated "Alternatively, if the S-coatomer interaction is abolished, virion fusogenicity is reduced while cell-cell fusion is increased due to greater anterograde S transport." In their assay, virion fusogenicity is decreased because the spike protein incorporation into the VLP seems to decrease. Therefore increasing the anterograde trafficking seems also to decrease the spike incorporation into VLP (however to a lesser extent) as seen when the retrograde transport is increased.

Reviewer #3 (Remarks to the Author):

This manuscript by Dey et al. describes the role of the SARS-CoV-2 Spike C-terminal, extra-membrane/cytosolic tail for the protein's trafficking within infected cells and for the balance between COP-mediated anterograde and retrograde transports, and towards retention for virus particle assembly and export to the cell surface. The authors use crystal structures of the COP-I subunits alpha and beta' WD40 domains in complex with S tail peptides and also WD40 mutants to rationalize the possible role of an acidic cluster for interaction with S. They find a lower affinity of WT S tail with beta'-WD40 than with alpha WD40 underlining the selective interactions also found for endogenous cargo. A clientized peptide, carrying an acidic mutation at the very C-terminus -as bona fide found in host cargo- enhances affinity and modifies the protein behavior in cell trafficking and vesicle formation.

The study exploits manifold binding data of WD40 and tail versions to rationalize selectivity from a structural perspective. NMR data on isotope labelled peptides are provided to compare WT and clientized tails alone and in complex with WD40 indicating that differences in solution conformations may play a role for complex formation. Finally, cellular reporter assays confirm findings on the role of S tail sequence invariability for the retention/cycling of S in cells and for the formation of VLPs. This study picks up earlier discussions on the role of cargo tail sequences in general, and in particular for CoVs. The structural data definitely add a strong value to the knowledge in the field and are relevant findings for the understanding of viral homeostasis. As correctly discussed by the authors, the robust CoV tail sequences may be well explained for the balanced re-sorting of S after infection. As such, the data may be of high relevance for vaccine design, but also for a general understanding of key regulatory aspects in viral particle assembly, which are as such not derivable from large Cryo-EM Spike structures so far. The work seems original to me seeing the first S tail structures in complex with COPs.

The provided data are trustworthy and convincing, but I do see some serious gaps. Especially I am not fully satisfied with the provided NMR data and the conclusions made from them. On top, the manuscript needs major additions in discussion of data and in some experimental descriptions, including inconsistencies. It reads well in terms of language, but a slightly more extensive introduction might be of help for a broad readership. Consequently, it needs to be revised before it may be considered for publication.

The authors are asked to address the following major issues:

- I wonder about binding rates and constants for WT proteins in BLI. Those curves are not shown in figure 2 for comparison. The same is true for table 1, which also does not include data for alpha-WD WT with short S tail peptide (non-amidated). What is the reason for the lack of these binding data? At least the complexes seem to crystallize. Have other binding assays been used instead?

Regarding NMR data:

- Seeing the broad interest, NMR chemical shifts need to be deposited in the BMRB.
- If conformational differences between WT S tail and the clientized version are suggested, what are the secondary chemical shifts (SCS) of the clientized peptide? Please explain the meaning of "extended strand-like structures" for the given residues! They also seem very weak. Can SCS be obtained in complex with WD40? Would the beta-propensity be yet more given in complex indicating a conformational selection? This would explain tiny differences in tail sequences to affect

complex formation on the dynamic/kinetic level as suggested by the authors in the discussion.

- What do the errors in Figure 4C stand for?

- Extended Data Figure S3: No peaks are labelled in panel A (other than stated in the legend). Has the clientized version been re-assigned or are peaks simply assigned by eye (which seems realistic)?

- All full spectra of tails titrated with WD40 domains should be provided. I do not understand what is shown in Ext. Data Fig. 5C, including the reason for highlighting Gly 1267 residue. What is the spectral overlay showing? Axis labeling is incorrect! Please check! What about the non-assigned residues in C? Please state in the legend.

- I am skeptical about the NMR ensembles of S tails, although the data are in principle highly appreciated and the experimental outcome interesting. Nevertheless, authors need to more precisely explain the experimental procedure of how ensembles have been generated. If I get it correct, the only restraint used is the PRE derived from one label position? Ensembles (as expected) do not converge very well (and probably not at all with side-chains depicted)? Authors should thus at least add RMSDs and the error (e.g. mean +/- STDEV) of the used distances to put the spotted difference in a statistical context. Although I do not see a proper explanation for the difference (except for charge repulsion) I do not generally doubt the finding and the possible conclusion as it may have a role for complex formation with COPs. However the basis for conclusions is simply based on one restraint (unless carbon SCS are e.g. different, see above) and should be treated with care. Else, authors may consider to add more data from the NMR side to corroborate their findings (a second PRE site, NOEs, backbone angles, CS).

- The text states that S-tails of Leu1244-Thr1273 have been used for GST pull-down experiments. Figure S1 does not show those sequences. Has the C-rich part been included? It needs to be clearer where exactly what part of the tail has been used as this varies between X-tal, NMR and other experiments. E.g., data in Ext. Figure S4 are said to be obtained from the fl-S-tails (line 202-203), but the sequence shown in panel A does not support this statement.

- I do not understand the middle section of page 6 explaining the alpha-beta'-chimera and the Tyr139Ala mutant. A graphical/structural depiction of this approach is needed to clarify the experimental meaning/relevance.

- Previous work (ref. 35) has shown that a Thr-to-Ala mutation at position 1273 also strongly influences the relative localization of S; it e.g. increases affinity to beta-COP-I. This would not be explainable with the contacts in the structures presented here. Has this S-tail mutant been tested? It may also influence dynamics/conformations of the tail. I see no need to perform experiments, but maybe this has been done (or other mutants been tested meanwhile)? Please comment at least!

- The conclusions in lines 300-302 on NMR data are weak in my eyes. I do not see a basis for extensive conformational changes of the tail by NMR. What does the statement refer to? Comparing WT and clientized tail, or apo and complexed tail? This needs clarification.

- Figure 5C implies different expression levels of protein versions (S0), esp. for number 4. Please comment!

In addition, I suggest to take care of the following minor, but relevant points:

- Please add more text describing the crystal structures, e.g. regarding the number of molecules in ASU. The SI should contain a full view of the domains with peptides.

- It may be interesting to (briefly) discuss how the complexes of WD40 with S-tail peptides compare to available structures of the COP-I subunits with known client motifs (e.g. PDB 2YNN), considering SARS-CoV-2 to exploit a unique binding mode.

- The manuscript text may be a bit clearer on where and why experiments have been carried out with alpha-WD40 vs. beta'-WD40. As far as I understand alpha-WD40 is more complicated to purify but a structure of the WT apo domain is available. How do they compare? And: it needs to be clearer that this study (as previous ones) uses yeast proteins, which seems established. How conserved are proteins and can findings easily be transferred between yeast (in vitro) and human (in cells)?
- Line 213 correlates Figure 4C and CSPs in the tail, but no CSPs are shown in this figure panel.
- The sentence in lines 295-297 needs revisions.
- The header of Table 1 appears wrong to me. Should it say: Binding rates (i.e. on- and off-rates) and dissociation constants of S tail peptides?
- I would appreciate a table of the constructs used in this study. C1253 residue is not included in any of the (overview) Figures, it is not easy to follow for readers. Also, including numbering of S-tail sequences would be of big help throughout the Figures.

REVIEWER COMMENTS

Reviewer#1

Comment 1.1: This manuscript by Hasan et al focuses on essentially the final two residues of the much-studied Coatomer-binding motif of SARS CoV2 S protein. Whilst well written and presented, and appears technically sound and exhaustive, it suffers greatly in its scope, which is narrow and of little interest to a general field as it largely reiterates what has been presented in many other studies. It doesn't really move the field along given all that has been published on it.

Response 1.1: We thank the reviewer for their positive comments while apologizing for the lack of clarity that the reviewer has experienced. We repeat here the novel aspects of our investigation and we have re-emphasized them in the manuscript-

- Our manuscript is the **first** to provide an atomic-level description of the spike C-terminal residue as a modulator of enhanced coatomer binding affinity. This moves the field forward because this C-terminal position makes direct contact with the coatomer and demonstrates large sequence diversity in endogenous and viral clients unlike the highly conserved dibasic motif. As such, the spike/client C-terminal residue functions as a key modulator of coatomer binding affinity and trafficking.
- Our manuscript is the **first** to describe affinity modulation provided by the WD40 basic cluster. This moves the field forward because the basic charge provided by this coatomer cluster in both aWD40 and b'WD40 domains strongly complements the acidic C-terminus often displayed by endogenous clients. As such, this discovery into the client C-terminus modulation of coatomer interactions has wide potential applications in the trafficking of several endogenous client proteins such those catalyzing drug detoxification and N-glycosylation.
- Our manuscript is the **first** to describe the structural and functional basis of partial mimicry wherein the CoV spike avoids host-like acidic residues at the C-terminus while displaying a host-like dibasic motif. This moves the field forward because it provides a key insight into the evolution of host factor hijacking towards coronavirus propagation. This work also defines the structural basis of the strong evolutionary pressure that maintains the spike C-terminus away from complete mimicry.
- Our manuscript is the **first** to provide insights into conformation of the spike cytosolic tail, which has remained an enigma despite the publication of more than 1200 spike structures. This moves the field forward because our data demonstrate conformational masking of coatomer-binding extended motif in the spike tail, for the **first time**. The details of long-distance conformational effects in trafficking regulation are presently poorly understood even in endogenous cellular clients.
- Our manuscript is the **first** to suggest a function of the spike acidic C-terminal residue in modulating accessibility of the dibasic motif. This is novel both in our approach of using solution-state NMR studies

on the spike tail both before and after WD40 binding. As the C-terminal position in client protein is often occupied by an acidic residue, this result provides broad and novel insights into local structural modulation of the dibasic motif *prior to* coatomer binding and trafficking.

- Our manuscript is the **first** to systematically compare the trafficking properties of the spike when expressed in isolation and when co-expressed with other coronavirus structural proteins for the wild-type and the clientized spike. This moves the field forward by providing comparative atomic to cellular level insights into spike trafficking when it is an isolated viral protein such as in a vaccine, and potential re-routing of the spike by other coronavirus proteins in infections.

The reviewer states that the data in our manuscript “*largely reiterates what has been presented in many other studies*”. It is not clear to us where such results have been published on structure-function insights into SARS-CoV-2 spike-coatomer interactions and modulation of spike trafficking by enhanced mimicry. The most prominent and relevant articles in the SARS-CoV-2 and coatomer fields that we are aware of are listed below.

- McBride CE, Li J, Machamer CE. The cytoplasmic tail of the severe acute respiratory syndrome coronavirus spike protein contains a novel endoplasmic reticulum retrieval signal that binds COPI and promotes interaction with membrane protein. *J Virol.* 2007 Mar;81(5):2418-28. doi:
- Jackson LP, Lewis M, Kent HM, Edeling MA, Evans PR, Duden R, Owen DJ. Molecular basis for recognition of dilysine trafficking motifs by COPI. *Dev Cell.* 2012 Dec 11;23(6):1255-62.
- Ma W, Goldberg J. Rules for the recognition of dilysine retrieval motifs by coatomer. *EMBO J.* 2013 Apr 3;32(7):926-37
- Jennings BC, Kornfeld S, Doray B. A weak COPI binding motif in the cytoplasmic tail of SARS-CoV-2 spike glycoprotein is necessary for its cleavage, glycosylation, and localization. *FEBS Lett.* 2021 Jul;595(13):1758-1767.
- Cattin-Ortolá J, Welch LG, Maslen SL, Papa G, James LC, Munro S. Sequences in the cytoplasmic tail of SARS-CoV-2 Spike facilitate expression at the cell surface and syncytia formation. *Nat Commun.* 2021 Sep 9;12(1):5333.
- Dey D, Singh S, Khan S, Martin M, Schnicker NJ, Gakhar L, Pierce BG, Hasan SS. An extended motif in the SARS-CoV-2 spike modulates binding and release of host coatomer in retrograde trafficking. *Commun Biol.* 2022 Feb 8;5(1):115.
- Li Y, Yang M, Nan Y, Wang J, Wang S, Cui D, Guo J, He P, Dai W, Zhou S, Zhang Y, Ma W. SARS-CoV-2 spike host cell surface exposure promoted by a COPI sorting inhibitor. *Acta Pharm Sin B.* 2023 Apr 18. doi: 10.1016/j.apsb.2023.04.007. Epub ahead of print. PMID: PMC10110937.

The information provided in these articles certainly provides the background for our present manuscript but they do not have any direct overlap with results on SARS-CoV-2 spike and coatomer interactions, other

than a few similar structural results on the wild-type spike tail and b'WD40 in the Li et al article that was published recently during the revision of the present manuscript.

Comment 1.2: There are several issues, which I think need addressing regardless of the outcome of review. No explanation is really given for why S binds alpha not beta' when its signal is closest to a beta' sequence. There is much discussion later on as to why many structure and modelling based experiments don't appear to work in respect to this and the authors predictions but no firm conclusions.

Response 1.2: We apologize if our writing was not clear on this point. Indeed, the principles of selectivity are complex, but we hope the reviewer finds the following summary helpful.

- **First:** The reviewer raises a **critical** point about the perceived resemblance of the spike tail sequence to that of other proteins that interact with b'WD40. To investigate this point, we compiled published data on client tail peptides and their selectivity for aWD40 or b'WD40 domains (new **Extended Data Table S5**). These data show an interesting trend wherein aWD40 demonstrates superior affinity for the client tail peptides from endogenous client proteins as well as for viral client proteins. We thank the reviewer for raising this intriguing point about selectivity, which we had previously not appreciated. We have now added this to our results. Please see:

“...The clientized S tail has broad selectivity for α WD40 and β 'WD40

Prior work revealed that the wild-type S heptapeptide does not bind to β 'WD40⁴¹, even though there is complete conservation of the basic cluster in α and β 'WD40^{61,62}. In fact, the α WD40 domain is favored by several client tails over the β 'WD40 domain...”

- **Second:** We have now added a more detailed discussion section on the complexity in client-WD40 selectivity:

“...We also elucidated the structural and biophysical basis of the balance between binding and dissociation of the S protein and coatomer. We showed that clientization of the S protein by mutation of Thr1273 to glutamate generates electrostatic complementarity between the acidic side chain of glutamate and a cluster of two basic residues conserved in α WD40 and β 'WD40 domains. The main chain carboxylate of Thr1273 and of Glu1273 interacts with Lys15, the third residue in this basic cluster. Site-directed mutagenesis of this basic cluster showed that an arginine at position 300 in α WD40 determines this enhanced affinity. In contrast, Arg13 and Lys15 residues in α WD40 provide the basis for basal binding to both wild-type and clientized spikes. Hence, the WD40 basic cluster has evolved distinct and fine-tuned interaction sites that modulate binding affinities for CoV S protein and client proteins. Our analysis thus reveals key atomic-level selectivity principles in coatomer-client interactions. Prior research has focused

mainly on two key differences in the WD40 domains, i.e., His31/Tyr139 in α WD40 are replaced by Tyr33/Phe142 in β 'WD40 domain, respectively^{43,61}. It has been inferred that His31 and Tyr139 provide favorable side-chain interactions for client tail binding that are absent from β 'WD40 domain. Furthermore, Tyr33 was suggested to be inhibitory to the binding of client tails to β 'WD40 domain wherein the penultimate tail residue is β -branched⁶¹. Building on these previous data, our present investigation shows that the principles of client-WD40 selectivity are more complex. In particular, the C-terminal acidic residue in client proteins modulates coatamer subunit binding affinity, and hence selectivity. In fact, clientization at the S tail C-terminus substantially overcomes selectivity against β 'WD40 domain. Moreover, β 'WD40 Tyr33 plays a critical role in stabilizing S tail binding through interactions with the tail Tyr1272 side-chain. Hence, we infer that this selectivity in favor of α WD40 and against β 'WD40 domain is not absolute but instead involves relative modulation of interaction affinity for one coatamer subunit versus the other through several interactions at the client-WD40 interface. Furthermore, our data suggest that simultaneous, high avidity engagement of α and β '-COPI subunits is likely driven by the C-terminal residue in endogenous oligomeric clients such as the ER-resident enzyme UGT⁶⁸, which is critical for xenobiotic processing and displays a dibasic motif in its cytosolic tail. Because the dibasic motif is displayed by a variety of proteins such as signaling proteins, enzymes, and growth factors⁴³, the structural principles of coatamer-client affinity modulation that we have elucidated have broad implications for secretory homeostasis...

Comment 1.3: Repeatedly throughout the authors state wt S does not bind beta' and no KD is given and yet it most clearly does bind or they could not have solved its structure – or is all this work based on an artefact? No explanation is given.

Response 1.3: We thank the reviewer for bringing this up. We agree that our original discussion on this issue was insufficient. We have added a new discussion section on this topic. Please see:

“... Our investigation addresses the fundamental basis of a popular approach in coatamer-client structural biology, in using β 'WD40 as a substitute for α WD40 for client tail peptide co-crystallization. We have recently reported that there is local structural similarity between an α WD40 loop and the N-terminal Lys residue in the client tail dibasic motif⁷¹. As such, the client tail peptides are often displaced by this α WD40 loop. Hence, although the α WD40 is favored by several clients in binding assays, it is often not suitable for co-crystallization with client peptides. Instead, β 'WD40 domain has been used as a substitute for α WD40 domain in co-crystallization even though this domain shows minimal binding to client tail peptides in biophysical approaches such as BLI and ITC. Our NMR analysis, which is substantially more suitable for assaying weak interactions, chemical shifts in S tail- β 'WD40 interactions that are weaker than those with α WD40 domain but are clearly

detectable. It is likely that high concentrations in crystallization push the equilibrium towards peptide- β 'WD40 binding, as suggested previously⁶¹. Hence, this analysis provides a biophysical basis for client tail co-crystallization with β 'WD40 domain..."

We expand on this topic here-

- **First:** Indeed, there is no detectable binding of b'WD40 to the wild type spike tail peptide in solution in BLI assays. We have published this analysis in Dey D, Singh et al, Commun Biol. 2022 Feb 8;5(1):115. These peptide-WD40 binding experiments are performed in the micromolar concentration range, which is consistent with the KD's of interaction with aWD40, the preferred binding partner of the spike tail. Interactions that are much weaker, such as of the spike tail peptide and the b'WD40 domain, will **not** be detected in this concentration range. However, the WD40 concentration is in the **molar** range in a crystal lattice as inferred from the unit cell constants and number of WD40 molecules per asymmetric unit (Extended Data Table S3). This is a **million-fold** higher concentration of b'WD40 than in the solution state binding experiments. Such high concentrations likely push the equilibrium towards spike tail peptide-b'WD40 binding in the crystal lattice. This is consistent with prior a publication from the Goldberg lab (Ma and Goldberg EMBO J 2013) wherein they showed that even though b'WD40 does not bind to dibasic peptides from several host and viral client proteins, peptide-b'WD40 complex formation is favored under crystallization conditions.
- **Second:** Although we would like to use higher b'WD40 concentrations in the BLI assays, high protein concentrations in the millimolar to molar range are known to cause non-specific secondary effects in techniques such as BLI and ITC. To demonstrate this, we present here a **BLI assay** wherein we went against the standard operating procedures and employed high concentrations of the analyte, b'WD40 (100 uM, 50 uM, and 25 uM; please see figure above), along with an immobilized clientized spike tail peptide (GVKLHYE). This BLI analysis shows clear pathological symptoms of non-specific crowding interactions and poor fitting of data to kinetic models.
- **Third:** NMR is one of the most appropriate techniques to investigate weak interactions that are beyond biophysical approaches such as BLI and ITC. Indeed, our NMR data show that there is interaction between the spike tail and the b'WD40 domain although this is **substantially** weaker than with aWD40 (**Extended Data Figure S7**). We had already included this in the original submission of our manuscript.

We have now emphasized this critical point in the revised manuscript in the paragraph cited above. Please see-

“...Our investigation addresses the fundamental basis of a popular approach in coatomer-client structural biology, in using β 'WD40 as a substitute for α WD40 for client tail peptide co-crystallization. We have recently reported that there is local structural similarity between an α WD40 loop and the N-terminal Lys residue in the client tail dibasic motif⁷². As such, the client tail peptides are often displaced by this α WD40 loop. Hence, although the α WD40 is favored by several clients in binding assays, it is often not suitable for co-crystallization with client peptides. Instead, β 'WD40 domain has been used as a substitute for α WD40 domain in co-crystallization even though this domain shows minimal binding to client tail peptides in biophysical approaches such as BLI and ITC. Our NMR analysis, which is substantially more suitable for assaying weak interactions, chemical shifts in S tail- β 'WD40 interactions that are weaker than those with α WD40 domain but are clearly detectable. It is likely that high concentrations in crystallization push the equilibrium towards peptide- β 'WD40 binding, as suggested previously⁶¹. Hence, this analysis provides a biophysical basis for client tail co-crystallization with β 'WD40 domain...”

- **Fourth:** To establish structural equivalence between the binding sites of α WD40 and β 'WD40 domains, we have generated a **new mutant Lys17Ala of β 'WD40**. This Lys17 residue is juxtaposed with Lys15 of α WD40. This mutation disrupts the basic cluster that interactions with the spike tail C-terminal carboxylate (GVKLHYE). This Lys17Ala mutation abolishes interactions of the β 'WD40 with the clientized spike tail peptide in a **new BLI assay (Figure 3G)**. This is consistent with a similar loss-of-binding effect seen in the juxtaposed Lys15Ala mutant in α WD40 (**Figure 2J-L**). Furthermore, this mutant demonstrates conformational rearrangement of a nearby Arg residue consistent with that seen in α WD40 Lys15Ala mutant (**Figure 3H**). This provides direct evidence of overlap and correspondence in the α WD40 and β 'WD40 binding sites for client tail peptides.

Comment 1.4: Indeed on line 151 it says a YtoA mutation structure was solved with S and three lines later it says Y to A abolishes binding to peptide – which or what is it?

Response 1.4: We apologize for the confusion. This entire section has now been rewritten. Please see:

“...Furthermore, our co-crystal structures showed that the Tyr33 side chain of β 'WD40 pushes away the Tyr1272 side chain of the S heptapeptide (Figure 3C)..... Moreover, this β 'WD40 Tyr33Ala mutation abolished binding to the clientized Thr1273Glu S heptapeptide (Figure 3F), suggesting a previously unrecognized role for β 'WD40 Tyr33 in S heptapeptide binding.”

To avoid confusion between co-crystallization and binding analyses, we request the reviewer to refer to Response 1.3 to the previous comment.

Comment 1.5: Why use such a short peptide for NMR PRE as this is unrepresentative of in vivo – use the whole cytoplasmic domain to make experiments valid.

Response 1.5 redacted:

Comment 1.6: Overall, I found the structural work of little general interest, focussing as it does on essentially one or two residues in a highly studied motif, although the cellular work between lines 78 and 100 was interesting, although again I failed to see the quite point of its importance.

Response 1.6: We thank the reviewer for correctly pointing out that two dibasic motif residues are extremely well characterized. That is precisely the reason that these dibasic residues are not the subject of our manuscript. Our focus is on the C-terminal position downstream of the dibasic motif in the spike tail. This is reflected in the title of our manuscript:

A single C-terminal residue controls SARS-CoV-2 spike trafficking and incorporation into VLPs

This position is unique in that it engages the coatamer not only through the side-chain but also through its main-chain carboxylate group, unlike the much studied dibasic motif which engages the coatamer through side-chain interactions. This position has not been studied in much detail, in stark contrast to the upstream dibasic motif.

The broad take home message of our work is: *The evolutionary selection against spike C-terminal residue mimicry is driven by electrostatic constraints and the need to disengage the coatamer at the virion assembly site in ERGIC. As such, the spike C-terminus has evolved into a critical affinity modulator that controls, (i) spike-coatamer dissociation, (ii) spike incorporation into progeny virions, and, (iii) cell-cell fusion for viral transmission.* The structural work provides the underlying atomic-level basis for this crucial finding.

In terms of more general interest such as for the wider field of secretory protein trafficking, we have provided a comparative structural analysis showing a high-degree of atomic level molecular mimicry of client peptides by the spike tail peptide. This shows that the very same WD40 basic cluster residues investigated here provide near identical interactions to the endogenous client tails. As such, coatamer affinity modulation by the C-terminal residue presented here is likely applicable to a wide number of endogenous client proteins. Hence, we anticipate that the structural, biophysical, and cellular principles derived in this investigation will be of broad and general interest to the trafficking research community.

To summarize the novelty of our work, our structural-biophysical study is the **first** to demonstrate the following-

- Conformational regulation of dibasic motif **prior to** coatomer binding. Previous studies have elucidated structural details of the tail **after** it binds the coatomer.
- Substitution of wild type Thr at this C-terminal spike position with an anionic Glu contributes directly to engagement of new interactions with a basic cluster on the coatomer WD40 surface
- Mutation of individual WD40 basic cluster residues yields a gradient of affinities in the mutant WD40 domain for the spike tail
- The Lys15Ala substitution in aWD40 and the juxtaposed Lys17Ala substitution in b'WD40 generate a large structural rearrangement of the WD40 binding site. This is highly relevant as an analogous mutant in b'WD40 has been employed in cellular studies of endogenous protein trafficking in yeast. Our result provides insights into the structural basis of this trafficking disruption (Jackson et al, Dev Cell 2012)
- Residues upstream of the spike tail dibasic motif modify coatomer interactions through conformational masking of the dibasic motif. This is consistent with cellular studies that suggest disruption of coatomer interactions upon Ala-scanning mutations in this upstream stretch in the spike tail (Cattin-Ortolá et al, Nat Commun 2021).
- The spike tail demonstrates conformational flexibility. Before our study, there was no structural information on spike tail conformations and the environment of the dibasic motif. In fact, there is barely any detailed information on the tail conformations of coatomer client proteins in general.

To further address reviewer#1's concern, we quote reviewer#3 verbatim:

*"...the data may be of high relevance for vaccine design, but also for a general understanding of key regulatory aspects in viral particle assembly, which are as such **not derivable from large Cryo-EM Spike structures so far**. The work seems original to me seeing the **first S tail structures in complex with COPs...**"*

We further quote reviewer#4 verbatim:

*"...The **real strength** of the current report is the inclusion of **atomic structural information** that is critical for understanding S-COPI interactions and function. Collectively, it provides insight into mechanisms involved in assembly of infectious particles and the role in fusion..."*

At the end, we thank the reviewer for raising several intriguing questions that have helped this manuscript and our own thinking on this topic. Modulation and regulation of biological interactions and pathways often depend on one or a few key protein residues. The evidence presented here supports such a critical role for the spike C-terminal residue, which has profound effects on coatomer binding, localization, virion incorporation, and fusogenicity. We have demonstrated this at multiple levels through orthogonal approaches: BLI with purified samples, co-crystallization, solution-state NMR, pull-down assays with cell

lysates, mass spectrometry, immunofluorescence imaging, protease digestion assays, VLP assembly assays, and cell culture fusion assays. The manuscript has been greatly modified to follow the reviewer's recommendations. We hope that the reviewer finds this information satisfactory and that the revised manuscript highlights the novelty and wide applicability of this work much more clearly than before.

Reviewer#2

Comment 2.1: The assembly mechanism of coronavirus assembly is still poorly characterized. The trafficking of the spike protein is dependent on its C-terminal tail. Retrograde trafficking of protein is mediated by the coatomer that recognizes dibasic residues at the C-terminal extremity of proteins (K-x-K-x-xCT or K-K-x-xCt) with the last residue being frequently acidic according to the authors. The spike protein contains a non canonical K-x-H motif that mediates its retrograde trafficking without any acidic residue at its C-terminus.

By analyzing sequences, the authors did not find any mutation of the last residue (T1273) into acidic residue that would increase the mimicry of the spike motif. The author mutated the Thr1273 into Glu and analyzed the effects of the mutation. They show an increase binding of the coatomer and an increased intracellular retention of the protein compared to the wild-type spike. The coatomer is composed of different subunits, among them the β -propellor WD40 domains of the α or β' subunit binds the retrieval signal on proteins, the spike protein has been shown to preferentially binds the WD40 domains. By using crystal structures and BLI assays using heptapeptide, the authors identified basic residue of the WD40 binding sites that are important for the increased binding of the spike motif bearing a T1273E mutation. Moreover, they show that this mutation allows the binding of the β' WD40 domains, meaning that this last residue is important for the spike-coatomer selectivity. They also characterized more precisely the interactions of S-WD40 domains and the importance of these interactions for WD40 domains selectivity. Then the authors analyzed the effect of the mutation in the context of the full-length C-terminal domain. By using NMR, they found that the C-tail of the wild-type protein may have a more compact conformation. They also confirmed in BLI assay the stronger binding of the C-tail harboring the T1273E mutation to the WD40 domain of the coatomer. Finally, they analyzed the effect of the mutation in cell-cell fusion assay and virus-like particle (VLP) production. The mutation T1273E decreased cell-cell fusion and incorporation of the spike protein in VLP.

These results are originals and provide information about the selectivity of the spike protein for the WD40 domain of the coatomer and more generally in term of cell biology and protein sorting. Moreover, they show that this non-optimal binding motif is important for the viral cycle and virus propagation. Indeed, optimization of the binding prevents spike incorporation into VLP and cell-cell fusion.

Response 2.1: We thank the reviewer for this thorough and well-composed summary.

Comment 2.2: Statistical analysis are missing

Response 2.2: All statistical analyses have now been explicitly listed. Please see below and in the manuscript:

- X-ray crystallography: We have explicitly listed the statistical quantities for resolution determination and for R-free calculation. Please see **Extended Data Table S3**.
- BLI assays: We have explicitly listed the accepted range of goodness-of-fit (R^2) and chi-squared analysis. Please see in the methods section:

“...All BLI sensorgrams reported in this investigation were within statistical limits, i.e., $R^2 > 0.98$ and $\chi^2 < 3...$ ”

- Pull-down assays: We have shown individual data points, mean, and, standard deviation of coatomer band intensity in Western blot analyses. Please see **Extended Data Figure S1**.
- PRE analyses: Individual measurements, mean, and standard deviations have been shown. Please see **Figure 4E**.
- Nano-luciferase luminescence assay and VLP fusogenicity assay: We have reported individual data points, mean, and standard error. Please see **Figures 5B, 5E, 6A, and 6D**.

Comment 2.3: Concerning the figure 5, for the cell-cell fusion assay, without careful reading of the material and methods, it is not clear that effector cells express all the structural proteins (M, E and HiBiT-N) and the conclusion line 230-231 does not make sense at first. This information should be added in the text or the legend of the figure.

Response 2.3: Thank you for pointing this out. We have added this information both in the main text-

“...In this assay, the full-length, 1273 residue S protein is on the PM of “effector” cells, which express one-half of a split luciferase along with M, E, and N-HiBiT...”

And the legend of **Figure 5**:

“...Schematic of S protein fusogenicity assay. If fusion-competent S proteins traffic to the surface of S-expressing “effector” HEK293T cells (co-express S, M, E, and N-HiBiT), they engage with “target” HEK293T cells expressing the human receptor for SARS-CoV-2 (hACE2) and activate cell-cell fusion....”

Comment 2.4: In the discussion line 261: the authors stated “Alternatively, if the S-coatomer interaction is abolished, virion fusogenicity is reduced while cell-cell fusion is increased due to greater anterograde S transport.” In their assay, virion fusogenicity is decreased because the spike protein incorporation into the VLP seems to decrease. Therefore increasing the anterograde trafficking seems also to decrease the spike incorporation into VLP (however to a lesser extent) as seen when the retrograde transport is increased.

Response 2.4: The reviewer raises an extremely important point, with which we completely agree. Please see the modified discussion-

“...Here, it is important to note that although S incorporation in VLPs is compromised both by substitutions that enhance S-coatomer binding and those that abolish coatomer binding, the underlying mechanism is likely different. Exclusive anterograde trafficking of S in the absence of coatomer-binding potentially reduces M-protein association, which is critical for S incorporation into VLPs. In contrast, enhanced coatomer-binding due to clientization likely prevents S release, thereby affecting S incorporation into virions...”

Reviewer #3 (Remarks to the Author):

Comment 3.1: This manuscript by Dey et al. describes the role of the SARS-CoV-2 Spike C-terminal, extra-membrane/cytosolic tail for the protein's trafficking within infected cells and for the balance between COP-mediated anterograde and retrograde transports, and towards retention for virus particle assembly and export to the cell surface. The authors use crystal structures of the COP-I subunits alpha and beta' WD40 domains in complex with S tail peptides and also WD40 mutants to rationalize the possible role of an acidic cluster for interaction with S. They find a lower affinity of WT S tail with beta'-WD40 than with alpha WD40 underlining the selective interactions also found for endogenous cargo. A clientized peptide, carrying an acidic mutation at the very C-terminus -as bona fide found in host cargo- enhances affinity and modifies the protein behavior in cell trafficking and vesicle formation.

The study exploits manifold binding data of WD40 and tail versions to rationalize selectivity from a structural perspective. NMR data on isotope labelled peptides are provided to compare WT and clientized tails alone and in complex with WD40 indicating that differences in solution conformations may play a role for complex formation. Finally, cellular reporter assays confirm findings on the role of S tail sequence invariability for the retention/cycling of S in cells and for the formation of VLPs.

This study picks up earlier discussions on the role of cargo tail sequences in general, and in particular for CoVs. The structural data definitely add a strong value to the knowledge in the field and are relevant findings for the understanding of viral homeostasis. As correctly discussed by the authors, the robust CoV tail sequences may be well explained for the balanced re-sorting of S after infection. As such, the data may be of high relevance for vaccine design, but also for a general understanding of key regulatory aspects in viral particle assembly, which are as such not derivable from large Cryo-EM Spike structures so far. The work seems original to me seeing the first S tail structures in complex with COPs.

Response 3.1: We thank the reviewer for their extremely positive assessment of our work.

Comment 3.2: The provided data are trustworthy and convincing, but I do see some serious gaps. Especially, I am not fully satisfied with the provided NMR data and the conclusions made from them. On top, the manuscript needs major additions in discussion of data and in some experimental descriptions, including inconsistencies. It reads well in terms of language, but a slightly more extensive introduction might be of help for a broad readership.

Response 3.3: We have expanded the introduction as suggested by the reviewer and included additional references to prior work on protein trafficking as well as to metabolic effects of dysfunction in trafficking, such as in cancers. Please see-

“... The α , β' , and ϵ subunits constitute the “B-subcomplex” whereas the β , δ , γ , and ζ subunits constitute the “F-subcomplex”, which shares significant structural similarity with the clathrin adaptors that function in PM trafficking³¹⁻³³. Overall, the coatomer functions to retrieve the escaped “client” proteins, such as type I membrane proteins, from cis-Golgi back to ER by retrograde trafficking. The coatomer binds to a C-terminal dibasic retrieval motif, Lys-x-Lys-x-x_{CT} or Lys-Lys-x-x_{CT} (x = any amino acid; x_{CT} = any C-terminal amino acid) in the client cytosolic tail, which leads to client packaging into COPI-coated vesicles and retrieval to ER³⁴⁻⁴⁰. The N-terminal β -propellor WD40 domains of B-subcomplex subunits, α and β' (α WD40 and β' WD40, respectively) provide binding sites for the dibasic motif on client proteins^{37,41-43}. Since the coatomer functions to retrieve a wide variety of eukaryotic proteins, it is not surprising that coatomer dysfunction has broad secretory effects and is linked to multiple disorders of growth, development, auto-immunity, and cancers^{24,43-54} ...”

Comment 3.4: I wonder about binding rates and constants for WT proteins in BLI. Those curves are not shown in figure 2 for comparison. The same is true for table 1, which also does not include data for alpha-WD WT with short S tail peptide (non-amidated). What is the reason for the lack of these binding data? At least the complexes seem to crystallize. Have other binding assays been used instead?

Response 3.4: We have previously published the BLI analysis of WT WD40 domains and their interactions to the spike peptides in Dey et al, Communications Biology, 2022. This article was cited in our manuscript. We have now added these data in **Table 1**. We apologize for this confusion and inconvenience.

Comment 3.5: Regarding NMR data: Seeing the broad interest, NMR chemical shifts need to be deposited in the BMRB.

Response 3.5: NMR chemical shifts have been deposited in the BMRB for the wild-type S tail, which has the C-terminal 21 amino acids of the spike protein, and for the Cys1253Ala/Thr1273Glu mutant (clientized) S tail (BMRB accession codes 51663 and 52007, respectively). This has been mentioned in the Data Availability section at the end of the manuscript.

Comment 3.6: If conformational differences between WT S tail and the clientized version are suggested, what are the secondary chemical shifts (SCS) of the clientized peptide?

Response 3.6: We have now ¹³C/¹⁵N-labeled the Cys1253Ala/Thr1273Glu mutant (clientized) S tail, assigned it by triple resonance methods, and performed secondary chemical shift analysis. Comparison of C α and CO

secondary chemical shifts for both the wild-type and mutant S tail peptides is shown in the new **Figure 4B**. Weak consensus β -propensity (consecutive negative $\Delta C\alpha$ and ΔCO values) is observed for N-terminal half residues Glu1258-Asp1260 in both the wild-type and mutant S tails. In contrast, the C-terminal half residues Val1268-His1271, which are in the dibasic motif WD40-binding region, have consensus β -propensity in the wild-type S tail but not the mutant. These data indicate that the Thr1273Glu mutation alters the conformational preferences of the binding epitope. This is further supported by the data in **Figure S3**, which shows that the Thr1273Glu mutation significantly affects the backbone amide chemical shift of Leu1270, located in the center of the dibasic motif.

Comment 3.7: Please explain the meaning of “extended strand-like structures” for the given residues! They also seem very weak.

Response 3.7: “Extended strand-like structures” refers to the β -propensity of residues that is inferred from secondary shift analysis, as described above. To improve clarity, we use “ β -propensity” in the revised manuscript. Based on comparison of secondary shifts for the wild-type and mutant S tails with typical corresponding values for folded proteins, we estimate ~10-20% population of these transient states.

Comment 3.8: Can SCS be obtained in complex with WD40? Would the beta-propensity be yet more given in complex indicating a conformational selection? This would explain tiny differences in tail sequences to affect complex formation on the dynamic/kinetic level as suggested by the authors in the discussion.

Response 3.8: ^{13}C -Assignments were not obtained for peptides in complex with WD40. However, based on our X-ray structure of the peptide/WD40 complex, we would expect the β -propensity of the region corresponding to the crystallographic epitope of the peptide to decrease in the bound state. Our data does indicate that the Thr1273Glu mutation leads to a decrease in consensus β -propensity in the binding region (**Figure 4B**), which might be interpreted to favor binding. However, we also note that the k_{on} rate is significantly slower for the mutant than the wild-type peptide (**Figure 4C, D**). The reasons for this are not completely clear but it appears that the Thr1273Glu mutation induces a change in conformational preference of the S tail that slows the kinetics of binding to WD40 domain. Based on our NMR data (**Figures 4B & S3**), we hypothesize that Glu1273 may form transiently stabilizing interactions with the sequentially nearest basic residue Lys1269. Having a proximal Lys1269-Glu1273 arrangement (even a transient one) would place the basic motif into a time-averaged state that is not as binding-competent as the wild-type, because these two residues are distal in the bound state. Once binding occurs, however, it is strongly stabilized

thermodynamically by favorable electrostatic interactions between Glu1273 and nearby basic residues on the WD40.

Comment 3.9: What do the errors in Figure 4C stand for?

Response 3.9: **Figure 4C** is now **Figure 4E**, which has been revised substantially. PRE experiments for wild-type and mutant S tails were each repeated multiple times (triplicate for the mutant, duplicate for the WT) and the results and statistics are summarized in the new **Figure 4E**. The PRE plots show the mean values and the error bars correspond with \pm one standard deviation.

Comment 3.10: Extended Data Figure S3: No peaks are labelled in panel A (other than stated in the legend). Has the clientized version been re-assigned or are peaks simply assigned by eye (which seems realistic)?

Response 3.10: The clientized (mutant) version was assigned by inspection in the original version of the manuscript. However, in order to obtain the secondary chemical shifts requested above, backbone resonances for the Thr1273Glu mutant have now been assigned using triple resonance NMR methods. Assignment labels have been added to **Figure S3A** for residues with the most notable CSPs upon mutation of Thr1273 to Glu1273 and a plot of the CSPs is shown in **Figure S3B**.

Comment 3.11: All full spectra of tails titrated with WD40 domains should be provided.

Response 3.11: Spectra showing the entire backbone amide region of bound S tails are now shown. The spectra of the unbound wild-type S tail overlaid with the α WD40-bound state are shown in the revised **Figure S4A**. The spectra of wild-type and Thr1273Glu mutant S tails overlaid with their β 'WD40-bound states are shown in **Figure S4B** and **Figure S4C**, respectively.

Comment 3.12: I do not understand what is shown in Ext. Data Fig. 5C, including the reason for highlighting Gly 1267 residue. What is the spectral overlay showing?

Response 3.12: This figure is replaced with **Figure S3**, which shows the effect of the Thr1273Glu mutation on the backbone amide chemical shifts of residues in the S tail dibasic motif. **Figure S3A** displays the overlaid two-dimensional ^1H - ^{15}N HSQC spectra of Cys1253Ala/Thr1273 and Cys1253Ala/Glu1273 S tails. **Figure S3B** is a plot of the backbone amide CSPs derived from panel A.

Comment 3.13: Axis labeling is incorrect! Please check!

Response 3.13: The axis labeling is now described more clearly. In all HSQC spectra, the x-axis corresponds with the ^1H dimension and the y-axis corresponds with the ^{15}N dimension.

Comment 3.14: What about the non-assigned residues in C? Please state in the legend.

Response 3.14: Extended data Figure S5C has been replaced with **Figure S3** as mentioned above. All backbone resonances in the wild-type and Thr1273Glu mutant S tails were assigned and the assignments have been deposited in the BMRB.

Comment 3.15: I am skeptical about the NMR ensembles of S tails, although the data are in principle highly appreciated and the experimental outcome interesting. Nevertheless, authors need to more precisely explain the experimental procedure of how ensembles have been generated. If I get it correct, the only restraint used is the PRE derived from one label position? Ensembles (as expected) do not converge very well (and probably not at all with side-chains depicted)?

Response 3.15: We agree with the reviewer and have decided to adopt a more conservative approach to the description of the S tail conformational ensemble. The PRE profiles for the wild-type and Thr1273Glu mutant S tails show a consistent and experimentally significant difference in the level of transient contacts between the N-terminal and C-terminal halves of the polypeptide chain. However, the differences are relatively small and difficult to capture reliably in structural representations of a flexible ensemble of conformers using a single PRE restraint. We therefore refer to the combined evidence provided by the PRE and CSP data in supporting our model of S tail conformation described as follows:

The polypeptide chain has a tendency to “fold back” on itself through transient contacts from the N-terminal Cys1255 to residues Val1264-His1271 in the C-terminal half of the S tail. This is true for both the wild-type and Thr1273Glu mutant S tail. However, the extent of these longer-range transient interactions is consistently attenuated in the mutant S tail. While the differences in PRE profiles are small, they are measurable and experimentally significant. **Our working model** is that electrostatic interactions between an acidic region near the N-terminus (Asp1257-Asp1260) and basic residues Lys1266 and Lys1269 in the C-terminal half are the primary drivers for these long-range transient contacts. When the C-terminal Thr1273 is mutated to Glu1273, this alters the charge balance, likely resulting in weakened interactions between the N-

and C-terminal halves of the S tail. Moreover, analysis of backbone amide chemical shift perturbations (CSPs) between the Cys1253Ala/Thr1273 and Cys1253Ala/Glu1273 S tail peptides (**Figure S3B**) demonstrates that mutation of the C-terminal residue from Thr1273 to Glu1273 has significant effects on residue Leu1270 in the center of the dibasic motif region. The non-neighboring (i-3) CSP is likely due to differences in transient contacts in the C-terminal half of the polypeptide chain brought about by the mutation (possibly a Lys1269-Glu1273 contact), which we postulate contribute to the differences seen in the PRE profiles of the wild-type and Thr1273Glu mutant S tails.

Comment 3.16: Authors should thus at least add RMSDs and the error (e.g. mean +/- STDEV) of the used distances to put the spotted difference in a statistical context.

Response 3.16: This has been done. Please see the legend of **Figure 4**.

...The plots show the mean values (bars, points) and error bars are for ± 1 standard deviation...

Source data have also been provided in excel files.

Comment 3.17: Although I do not see a proper explanation for the difference (except for charge repulsion) I do not generally doubt the finding and the possible conclusion as it may have a role for complex formation with COPs. However the basis for conclusions is simply based on one restraint (unless carbon SCS are e.g. different, see above) and should be treated with care.

Response 3.17: In addition to obtaining PRE profiles with improved statistics (**Figure 4E**), we now also have more detailed analysis from $^{13}\text{C}\alpha$ and ^{13}CO secondary shifts (**Figure 4B**) and from CSP comparisons between the wild-type and Thr1273Glu mutant S tails (**Figure S3**). All three of these datasets point to differences in the C-terminal region around the WD40-binding motif. Nevertheless, we do agree with the reviewer and present a conservative model for the interactions in the wild-type and Thr1273Glu mutant S tails as described above that is consistent with our experimental observations.

Comment 3.18: Else, authors may consider to add more data from the NMR side to corroborate their findings (a second PRE site, NOEs, backbone angles, CS).

Response 3.18: Additional NMR chemical shift data has been added to our analysis (see above). We also did consider using a second PRE site. From the point of view of reciprocal interaction, the best place for a second

spin label would be at the C-terminus. However, this is where the long-range conformational differences between the wild-type and mutant spike tails are detected. Moreover, introducing a Cys mutant in the C-terminus would likely be perturbing and may provide confounding results. Alternatively, placing a spin label near the center of the peptide will have approximately equal effects over the ~10 amino acid range in both directions and not be useful in detecting conformational differences. Thus, we concluded that neither of these approaches would enable us to discern differences between the wild-type and mutant conformational ensembles beyond what we have already described in the manuscript.

Comment 3.19: The text states that S-tails of Leu1244-Thr1273 have been used for GST pull-down experiments. Figure S1 does not show those sequences. Has the C-rich part been included? It needs to be clearer where exactly what part of the tail has been used as this varies between X-tal, NMR and other experiments. E.g., data in Ext. Figure S4 are said to be obtained from the fl-S-tails (line 202-203), but the sequence shown in panel A does not support this statement.

Response 3.19: We thank the reviewer for highlighting the need to summarize the spike constructs used in this investigation. This has now been provided in **Extended Data Table S1**. Briefly, we included the Cys-enriched portion of the tail in our GST constructs to ensure consistency with previous publications in this field (Jennings et al, FEBS L, 2021 and Cattin-Ortolá et al, Nat Commun 2021) and to provide a linker between bulky GST and the spike tail.

Comment 3.20: I do not understand the middle section of page 6 explaining the alpha-beta'-chimera and the Tyr139Ala mutant. A graphical/structural depiction of this approach is needed to clarify the experimental meaning/relevance.

Response 3.20: We apologize for the confusion caused by this section. We have now simplified this section and refocused it exclusively on the b'WD40 domain rather than on multiple constructs of b'WD40 and aWD40 domains. Please see:

“...The enhanced interaction of the clientized S tail heptapeptide provided us with a tool to test the validity of β 'WD40 domain in our structural experiments...”

Comment 3.21: Previous work (ref. 35) has shown that a Thr-to-Ala mutation at position 1273 also strongly influences the relative localization of S; it e.g. increases affinity to beta-COP-I. This would not be explainable with the contacts in the structures presented here. Has this S-tail mutant been tested? It may also influence

dynamics/conformations of the tail. I see no need to perform experiments, but maybe this has been done (or other mutants been tested meanwhile)? Please comment at least!

Response 3.21: We thank the reviewer for raising this point. We reported this BLI experiment in our previous publication, Dey et al Communications Biology 2022. We observed a modest enhancement of WD40 binding in BLI assays with the mutant spike heptapeptide, GVKLHYA. In view of this previous publication and the present structural data, it seems more likely that this Thr1273Ala mutation causes conformational changes in the spike tail that indirectly enhance affinity for the coatomer. We have now added this to the discussion (please see...)

“...In contrast, the Thr1273Ala substitution likely modifies the S tail conformational accessibility for coatomer binding, which results in its enhanced retention in early secretory compartments⁵³...”

Comment 3.22: The conclusions in lines 300-302 on NMR data are weak in my eyes. I do not see a basis for extensive conformational changes of the tail by NMR. What does the statement refer to? Comparing WT and clientized tail, or apo and complexed tail? This needs clarification.

Response 3.22: This entire section has now been rewritten in view of the new NMR data. Please see-

“...Although retrograde and anterograde trafficking of the S protein are critical for SARS-CoV-2 propagation, the mechanism underlying trafficking bifurcation at the cis-Golgi is not well understood. Structural insights from our NMR studies advance the understanding of this intriguing mechanism. S molecules in the cis-Golgi with an inaccessible extended motif will be incompetent for coatomer binding and continue anterograde export to the PM. In contrast, an accessible extended motif will permit S retrieval to the virion assembly site in ERGIC. The NMR data presented here suggest a turn-like structure of the wild-type S tail, which can potentially present conformational constraints on coatomer interactions. Furthermore, the inherent conformational flexibility in the S tail likely enables exposure of the extended motif for coatomer binding. Such an equilibrium in S tail states might be modulated by factors such as oligomerization, palmitoylation, proximity to the Golgi membrane, association with cytosolic proteins, and mutations outside the extended motif^{13,69,70}. For example, mutagenesis of the SARS-CoV-2 S tail has identified residues upstream of the extended motif (Ser1261-Glu1262) that are critical for coatomer interactions¹³. Our structural analysis maps these two residues to a turn in the S tail, suggesting that the Ser1261Ala/Glu1262Ala double mutation modifies the conformational accessibility of the extended motif and reduces coatomer binding. In contrast, the Thr1273Ala substitution likely modifies the S tail conformational accessibility for coatomer binding, which results in its enhanced retention in early secretory compartments¹³. On a local level, our data show a function of an acidic residue at the S protein C-terminus in modulating dibasic motif accessibility. Given that several client proteins display acidic residues at the C-terminal position, our data are suggestive of a broad modulatory function of this residue in secretory trafficking. Overall, our analysis reveals that indirect, long-range conformational

dynamics as well as local interactions determine S-coatomer binding and the balance of anterograde-retrograde trafficking...

Comment 3.23: Figure 5C implies different expression levels of protein versions (S0), esp. for number 4. Please comment!

Response 3.23: This was due to a loading error in the gel. This figure has now been corrected. We apologize for this confusion.

Comment 3.24: Please add more text describing the crystal structures, e.g. regarding the number of molecules in ASU. The SI should contain a full view of the domains with peptides.

Response 3.24: We have added this information. Please see:

“...Overall, the two crystal structures and reveal substantial similarity ($C\alpha$ RMSD=0.2Å). The S tail heptapeptides are intercalated between two parallel β' WD40. The tail extended motif makes contact with the binding site residues in one of the two β' WD40 domains whereas the upstream tail heptapeptide residues interact with the opposite face of the other β' WD40 domain...”

Please also see the new Extended Data Figure S3 which provides full views of the WD40 domains with the peptides, and Extended Table S3 that contains information on molecules in ASU.

Comment 3.25: It may be interesting to (briefly) discuss how the complexes of WD40 with S-tail peptides compare to available structures of the COP-I subunits with known client motifs (e.g. PDB 2YNN), considering SARS-CoV-2 to exploit a unique binding mode.

Response 3.25: We have added this new text along with Extended Data Table S4. Please see:

“...Furthermore, the S tail heptapeptides demonstrate substantial structural similarity with previously published co-crystal structures of β' WD40 domain with the tail peptides from endogenous client proteins and a porcine CoV S protein...”

Comment 3.26: The manuscript text may be a bit clearer on where and why experiments have been carried out with alpha-WD40 vs. beta'-WD40. As far as I understand alpha-WD40 is more complicated to purify but a

structure of the WT apo domain is available. How do they compare? And: it needs to be clearer that this study (as previous ones) uses yeast proteins, which seems established. How conserved are proteins and can findings easily be transferred between yeast (in vitro) and human (in cells)?

Response 3.26: The reviewer is correct about challenges in aWD40 purification. We have added a new section for clarification:

“...We used WD40 domains from the yeast α -COPI and β '-COPI subunits, which have been used extensively in prior structural investigations^{41,61,62}. The binding site residues are 100% identical between yeast and human α WD40 domains (PDB IDs 7S22 and 6PBG). The yeast and human β 'WD40 domain binding sites differ at one residue, i.e., Phe142 in yeast β 'WD40 is replaced by Tyr144 in human β 'WD40 (PDB ID 8D30). Contacts between symmetry-related α WD40 molecules occluded the binding site in these crystals and no peptide density was observed. Hence, we subsequently adopted a previously described strategy of using β 'WD40 for co-crystallization of peptides containing the Lys-x-His-x-xCT motif.”

During the revision of this manuscript, we published a new manuscript on the challenges of producing aWD40 for structural biology:

Dey D, Hasan SS. Strategies for rapid production of crystallization quality coatomer WD40 domains. Protein Expr Purif. 2023 Dec;212:106358. doi: 10.1016/j.pep.2023.106358. Epub 2023 Aug 23. PMID: 37625737.

Comment 3.27: Line 213 correlates Figure 4C and CSPs in the tail, but no CSPs are shown in this figure panel.

Response 3.27: Line 213 refers to **Figure S4** only. This has been corrected in the manuscript.

Comment 3.28: The sentence in lines 295-297 needs revisions.

Response 3.28: We have rephrased this sentence. Please see:

“...This leads to an interesting hypothesis, i.e., early in infection, the coatomer sequesters the S protein near ERGIC to provide a window for the assembly of infectious progeny virions, prior to S export to PM at a later stage for cell-cell fusion and virion transmission...”

Comment 3.29: The header of Table 1 appears wrong to me. Should it say: Binding rates (i.e. on- and off-rates) and dissociation constants of S tail peptides?

Response 3.29: This change has been made. Please see-

Table 1: Binding rates and dissociation constants of spike tail peptides interacting with coatomer WD40 domains

Comment 3.30: I would appreciate a table of the constructs used in this study. C1253 residue is not included in any of the (overview) Figures, it is not easy to follow for readers. Also, including numbering of S-tail sequences would be of big help throughout the Figures.

Response 3.30: A table of all the S construct sequences in this manuscript has been added in Extended Data Table S1.

Reviewer#4

Comment 4.1: The Dey et al. manuscript focuses on the roles of carboxy terminal residues in the SARS-CoV-2 spike (S) protein in virus assembly and fusion. Much earlier studies with multiple coronavirus S proteins showed that the proteins contain a dibasic motif in the cytoplasmic tail that is similar to ER retrieval signals and that the motif is important for S transport. Further, it was shown that the motif binds COPI. The current report builds off the earlier studies and extends our current understanding of S bidirectional trafficking and transport to the cell surface. The real strength of the current report is the inclusion of atomic structural information that is critical for understanding S-COPI interactions and function. Collectively, it provides insight into mechanisms involved in assembly of infectious particles and the role in fusion.

Response 4.1: We thank the reviewer for this well-composed summary and for identifying the strengths of this manuscript.

Comment 4.2: Fig. 1A – The model is nice and important to be included as Fig. 1A but, it needs to be modified and simplified. Steps 2 (export vesicles from ER) and 3 (trafficking to ERGIC) are switched. Step 4 arrow should go directly to cis-Golgi. Step 4' is fine to illustrate direct transport to the cell surface. Step 5 encompasses the COPI vesicles retrieval to ER& ERGIC. Vesicles pinching off the cis-Golgi is clear so no need for numbering.

Response 4.2: Thank you for pointing out these limitations. We have made the suggested changes and simplified the figure by using arrows with unambiguous positioning.

Comment 4.3: Fig. 2A, lines 111-114 – Thr1273 also interacts with Arg15?

Response 4.3: Yes, the Arg15 side-chain interacts with the main-chain carboxylate group of the S C-terminus.

Comment 4.4: Page 9, lines 261-263 – Virion particle assembly or VLPs are not necessarily reduced. The amount of S incorporated into particles is reduced, but M-E-N or M-E particles will presumably continue to be made. This is not measured, but the amount of N appears to be the about the same in the VLP experiments even when S is reduced. In the context of virus infection, the amount of infectious virus will be reduced. This is implicit, but good to clarify for readers that do not think about or know much about assembly.

Response 4.4: We completely agree and thank the reviewer for this suggestion. In response, we have modified the title of the manuscript:

*A single C-terminal residue controls SARS-CoV-2 spike trafficking and virion assembly*

To-

*A single C-terminal residue controls SARS-CoV-2 spike trafficking and VLP incorporation*

Furthermore, we have added the following sentence to the discussion to clarify that point:

“...It must be noted that clientization does not interfere with the biosynthesis of the S protein, but rather exerts its effect by modifying S localization, fusogenic activity, and VLP incorporation. As such, a single residue at the C-terminus of the S tail is a critical determinant of SARS-CoV-2 propagation and infectivity....”

Comment 4.5: Since there are multiple “clientized” S proteins used in the study, it would be helpful to include in Fig. 1 a sub-panel showing the alignment of the last few WT and clientized S residues for easy reader reference. An alignment like the one included Fig. S1 would be useful in the main body of the manuscript. It would also be helpful to include the residues in the labels in Fig. 1D, E, F, rather than labeling just as clientized S. Include Thr1273Glu, etc. as in Figs. 5 and 6 labels.

Response 4.5: Thank you for your suggestion and we apologize for not including a list of sequences in the original draft. In response, we have added sequence information at several places in the manuscript-

- Figure 1A, lower panel
- Extended Data Table S1
- Throughout Figures 1-6 and Extended Data Figures, wherever clientization is mentioned
- Throughout the text wherever clientization is mentioned

Comment 4.6: Referral of the mutant proteins as the clients is not the best designation. Why not just simplify and refer to the mutants directly as Thr172Glu, etc. Even some of the WT 2 molecules are a “client” for retrieval by coatamer.

Response 4.6: We feel that the use of “clientization” conveys the rationale behind the Thr1273Glu to Thr1273Asp substitution. This also distinguishes these spike constructs from naturally occurring mutations. Hence, we humbly continue using “clientization” in this manuscript. However, we confess that we have considered alternatives to clientize such as, “hostize” and “enhanced mimicry substitute”.

Comment 4.7: Minor point -Label HiBiT-N in Fig. 6B.

Response 4.7: This correction has been made.

REVIEWERS' COMMENTS

Reviewer #1 (Remarks to the Author):

The manuscript has been somewhat improved by the additional explanations and is now suitable for publication. However its scope, general interest and constant use of a proxy of beta' instead of the reported bona fide partners alpha points to it being more suitable for publication in Science Reports rather than Nat Comm, which is high impact and regularly publishes cutting edge research - this is not cutting edge but solid and staid

Reviewer #3 (Remarks to the Author):

The authors have substantially improved the quality of the manuscript and addressed all raised concerns to my satisfaction. NMR data are now treated more thoroughly and are also not overinterpreted.

The manuscript also appears more structured in its discussion, the reference to earlier work by the authors, and the depiction of data incl. statistics.

I do recommend the manuscript for acceptance and only want to mention some minor issues (no need to see for me again):

- in all NMR 2D plots I suggest to add p.p.m. as axis units (actually also delta as the CS)
- Fig. 4E should have a y-axis label, e.g. (Iox/Ired)
- Fig. S1a: add MW values from standard for orientation
- please use consistent labeling of y-axes in Rluc assays, e.g. for luminescence values between Fig. 5b and 6d

Reviewer #1 (Remarks to the Author):

Comment: The manuscript has been somewhat improved by the additional explanations and is now suitable for publication. However its scope, general interest and constant use of a proxy of beta' instead of the reported bona fide partners alpha points to it being more suitable for publication in Science Reports rather than Nat Comm, which is high impact and regularly publishes cutting edge research - this is not cutting edge but solid and staid

Response: We are thankful to the reviewer for considering the improvements to the manuscript. We hope to use the reviewer's suggestions on alpha-WD40 and beta'-WD40 in our next publications.

Reviewer #3 (Remarks to the Author):

Comment: The authors have substantially improved the quality of the manuscript and addressed all raised concerns to my satisfaction. NMR data are now treated more thoroughly and are also not overinterpreted. The manuscript also appears more structured in its discussion, the references to earlier work by the authors, and the depiction of data incl. statistics.

Response: We thank the reviewer for the positive assessment of our revised manuscript.

Comment: I do recommend the manuscript for acceptance and only want to mention some minor issues (no need to see for me again):

- in all NMR 2D plots I suggest to add p.p.m. as axis units (actually also delta as the CS)

Response: This has been done. Please see-

Figure 4A:

Supplementary Figure S6-

Supplementary Figure S7-

- Fig. 4E should have a y-axis label, e.g. (lox/lred)

Response: This has been done. Please see-

- Fig. S1a: add MW values from standard for orientation

Response: This has been done. Please see-

- please use consistent labeling of y-axes in Rluc assays, e.g. for luminescence values between Fig. 5b and 6d

Response: This has been done using the units, "RLU". Please see-

Figure 5B-

Figure 6D-